# Uncovering natural variation in root system architecture and growth dynamics using a robotics-assisted phenomics platform

Therese LaRue[1,2†], Heike Lindner[2,3†], Ankit Srinivas[2], Moises Exposito-Alonso[1,2], Guillaume Lobet[4,5], José R Dinneny[1]*

[1]Department of Biology, Stanford University, Stanford, United States; [2]Department of Plant Biology, Carnegie Institution for Science, Stanford, United States; [3]Institute of Plant Sciences, University of Bern, Bern, Switzerland; [4]UCLouvain, Faculty of Bioengineering, Louvain-la-Neuve, Belgium; [5]Forschungszentrum Jülich, Agrosphere Institute, Juelich, Germany

**Abstract** The plant kingdom contains a stunning array of complex morphologies easily observed above-ground, but more challenging to visualize below-ground. Understanding the magnitude of diversity in root distribution within the soil, termed root system architecture (RSA), is fundamental in determining how this trait contributes to species adaptation in local environments. Roots are the interface between the soil environment and the shoot system and therefore play a key role in anchorage, resource uptake, and stress resilience. Previously, we presented the GLO-Roots (Growth and Luminescence Observatory for Roots) system to study the RSA of soil-grown *Arabidopsis thaliana* plants from germination to maturity (Rellán-Álvarez et al., 2015). In this study, we present the automation of GLO-Roots using robotics and the development of image analysis pipelines in order to examine the temporal dynamic regulation of RSA and the broader natural variation of RSA in *Arabidopsis*, over time. These datasets describe the developmental dynamics of two independent panels of accessions and reveal highly complex and polygenic RSA traits that show significant correlation with climate variables of the accessions' respective origins.

**\*For correspondence:** dinneny@stanford.edu

[†]These authors contributed equally to this work

**Competing interest:** The authors declare that no competing interests exist.

## Editor's evaluation

The authors present an automated system for phenotyping root system architecture based on bioluminescent roots resulting from a constitutively expressed luciferase transgene (GLO-Roots). They have developed a robotics-assisted phenotyping platform and an automated image analysis pipeline for high throughput analysis. An impressive array of 93 luciferase expressing *Arabidopsis thaliana* accessions provides a major resource for understanding the genetic basis for root system architecture variation under physiologically relevant conditions. The work will be of great interest to plant biologists and all those studying genetic variation in plants.

## Introduction

The diversity of shoot and root system forms within the plant kingdom mirrors the many functions required for plant survival in diverse habitats. However, these forms are not static, but dynamically change during the lifespan of a plant as it acclimates to a changing environment, and over evolutionary time scales as a consequence of natural selection and genetic drift. The visible above-ground

shoot system is optimized for photosynthesis and reproduction while the below-ground root system is responsible for anchoring the plant and allowing it to reach essential nutrients and water, which is mediated, in part, through interactions with microbes in the soil (*Lundberg et al., 2012*; *Bulgarelli et al., 2012*; *Castrillo et al., 2017*; *Harbort et al., 2020*). While shoot forms and structures have undergone centuries of scrutiny under the human eye, underground root systems have received much less attention due to impeded accessibility.

Root system architecture (RSA) refers to the spatial arrangement of roots in soil, which encompasses the geometric nature of root connectivity and distribution (*Lynch, 1995*; *Rellán-Álvarez et al., 2016*). This arrangement of roots depends on relationships among the growth rates, branching frequency, and gravity set point of different root types. In allorhizic root systems, common in Eudicotyledons like the model plant *A. thaliana*, the embryonically formed primary root is the first to emerge from the seed at germination and ultimately branches to give rise to secondary roots, which, in turn, branch and produce tertiary and higher order roots (*Osmont et al., 2007*; *Fitter, 2017*). Each step of forming a branch, from the priming of a lateral root primordia to emergence from the parent root, is a tightly controlled process regulated by endogenous and exogenous signals (*Moreno-Risueno et al., 2010*; *Péret et al., 2009*; *Van Norman et al., 2013*; *Motte et al., 2019*). Modulation and crosstalk of plant hormones change local growth patterns. The ratios of the phytohormone auxin and its antagonist, cytokinin, control growth throughout the plant and, importantly, regulate primary and lateral root initiation (*Tian et al., 2014*; *Morris et al., 2017*). The hormone abscisic acid (ABA), on the other hand, inhibits lateral root primordia from emerging (*De Smet et al., 2003*). ABA levels can increase in response to external stress and cause an extended quiescent phase after lateral root emergence (*Duan et al., 2013*; *Xiong et al., 2006*). Hence, the precise balance of these hormones influences root development and shapes overall RSA.

In addition to their genetically determined growth program, plants also react to spatial and environmental cues to tune their form for certain environments. Roots must strategically navigate through the soil environment for efficient acquisition of water and nutrients (*Rellán-Álvarez et al., 2016*; *Dinneny, 2019*). Understanding the mechanisms by which roots sense and respond to their environment remains a significant challenge since existing methods either compromise on physiological relevance or on throughput, which can impact the ability to deploy functional genomic tools to identify regulatory genes. Many RSA studies are performed on young root systems using easily accessible, in vitro gelmedia-based systems and leverage the powerful genetic and molecular resources of *A. thaliana* (*Rosas et al., 2013*; *Ogura et al., 2019*; *Waidmann et al., 2019*; *Julkowska et al., 2014*). Whereas bigger and more mature root systems of crop species have been studied in soil environments, however, compromising on throughput (*Morris et al., 2017*; *Jiang et al., 2010*).

Previously, we developed GLO-Roots (Growth and Luminescence Observatory for Roots), an imaging platform enabling visualization of soil-grown *Arabidopsis* root systems from germination to maturation by combining custom growth vessels, luminescence-based reporters, an imaging system, and an image analysis suite (*Rellán-Álvarez et al., 2015*). Here, we present the automation of GLO-Roots by creating GLO-Bot, a robotic platform that enables large-scale capture of *Arabidopsis* root growth over time in a soil-like environment. Along with the automated imaging system, we developed an improved root analysis pipeline for quantification of various RSA traits. Together, these technological advancements enabled time-lapse imaging of RSA development of 93 *Arabidopsis* accessions from 14 to 28 days after sowing, creating a unique dataset that allows for wide-ranging analyses of highly complex, adult root systems. Quantitative genetic analyses and genome-wide association studies (GWAS) yielded insight into the heritability and genetic architecture of distinct RSA traits, and significant associations with climates across the species' geographic distribution highlighted the power of our approach and its relevance for understanding how root architectural traits contribute to local adaptation.

## Results

### Automation enables high spatio-temporal resolution measurements of root system dynamics

#### Establishing the robotics infrastructure for automated root imaging

To further develop the potential of GLO-Roots for time-lapse imaging of root development, from early stages to mature RSAs, we developed GLO-Bot, a Cartesian gantry robotic system consisting of an exterior frame and arm on a linear rail system built from T-Slotted aluminum extrusion, which houses the Growth and Luminescence Observatory 1 (GLO1) imaging system (*Rellán-Álvarez et al., 2015*; *Figure 1A and B*). In addition to GLO1, the frame contains a growth area for seven black polyethylene growth boxes (*Figure 1B*), each of which holds 12 rhizotrons (also known as rhizoboxes) arranged in a two by six grid (*Figure 1C*, *Figure 1—figure supplement 1*). In total the system can hold 84 rhizotrons, which typically contain one plant.

The primary goal of developing the robotics platform was to automate plant imaging since the luciferin watering regime of each rhizotron and the long exposure time required to capture luminescence signal were the most time-consuming steps of the GLO-Roots system. An important consideration was the layout for the robot and range of motion for the arm (*Figure 1A*). The perpendicular position of the two cameras relative to the rhizotron being imaged in GLO1 lent itself well to the final linear rail system and arm with x-, y-, and z-motors. To minimize positional errors associated with finding a particular rhizotron, the robotic arm checks the coordinates of each growth container prior to imaging (*Video 1*). The arm moves to each plant position and lifts the rhizotron via a metal hook, which gives the arm a robust surface to pick up the rhizotron (*Figure 1C*, *Figure 1—figure supplement 1A*). Both rhizotron and growth box designs were adapted for robotic use in a way that prevents light exposure of the root system during growth (*Shi et al., 2018*) while also facilitating easy removal and repositioning of rhizotrons in the growth container during imaging. Each rhizotron is shielded from light exposure by creating individual chambers within the growing box (*Figure 1C*, *Figure 1—figure supplement 1B*). To facilitate removal and replacement of each rhizotron for imaging, we changed the design of the growth box from a standing to a hanging position by adding a piece of acrylic at the top of each rhizotron, which, with the help of small guide pegs, allows the rhizotron to hang in its proper position and shields light from entering the growth container (*Figure 1C*).

Programed movements of the robotic arm bring the rhizotron to a barcode scanner, which allows the system to recall the specific treatment and imaging protocol to use (*Video 1*). Subsequently, the arm brings the rhizotron to a watering station where a luciferin solution is applied prior to imaging, which provides the substrate for the constitutively expressed luciferase (*Rellán-Álvarez et al., 2015*; *Figure 1A*; *Figure 2*). Slow back and forth movements of the rhizotron under a pipette tip connected to a peristaltic pump provide the solution ( *Video 1*). As the arm moves, it bypasses a specified gap in the center of the rhizotron to avoid contact with the shoot. The entire watering process takes approximately 10 min, which allows the water to slowly seep through and spread throughout the rhizotron. The automation was set up such that while one plant is being imaged, the next plant is being watered. To minimize arm movement time, watered rhizotrons are placed on a ledge while the arm removes the imaged rhizotron from the imager and returns it to its growth box (*Video 1*). If the system is running at maximum capacity, GLO-Bot can process 96 rhizotrons in 24 hr, by growing an eighth box on a nearby shelf and swapping out one of the seven boxes in the system during the day. However, the following experiments were all imaged at approximately half capacity in order to respond to possible technical errors.

### Development of an image analysis pipeline for dynamic trait quantification

#### Image alignment, signal optimization, and background noise removal

The increased throughput of GLO-Bot enabled time-lapse imaging of root systems and created new opportunities for the measurement of dynamic root growth traits, but also revealed new challenges for root image analysis. Luciferase signal intensity varies between transgenic LUC-expressing lines and decreases in older root tissues, making it even more difficult to distinguish between roots and noise during the course of the experiment (*Figure 2*). While this signal variation is problematic for analysis of raw images, we leveraged our time-lapse data to capture a strong signal for the entire root system

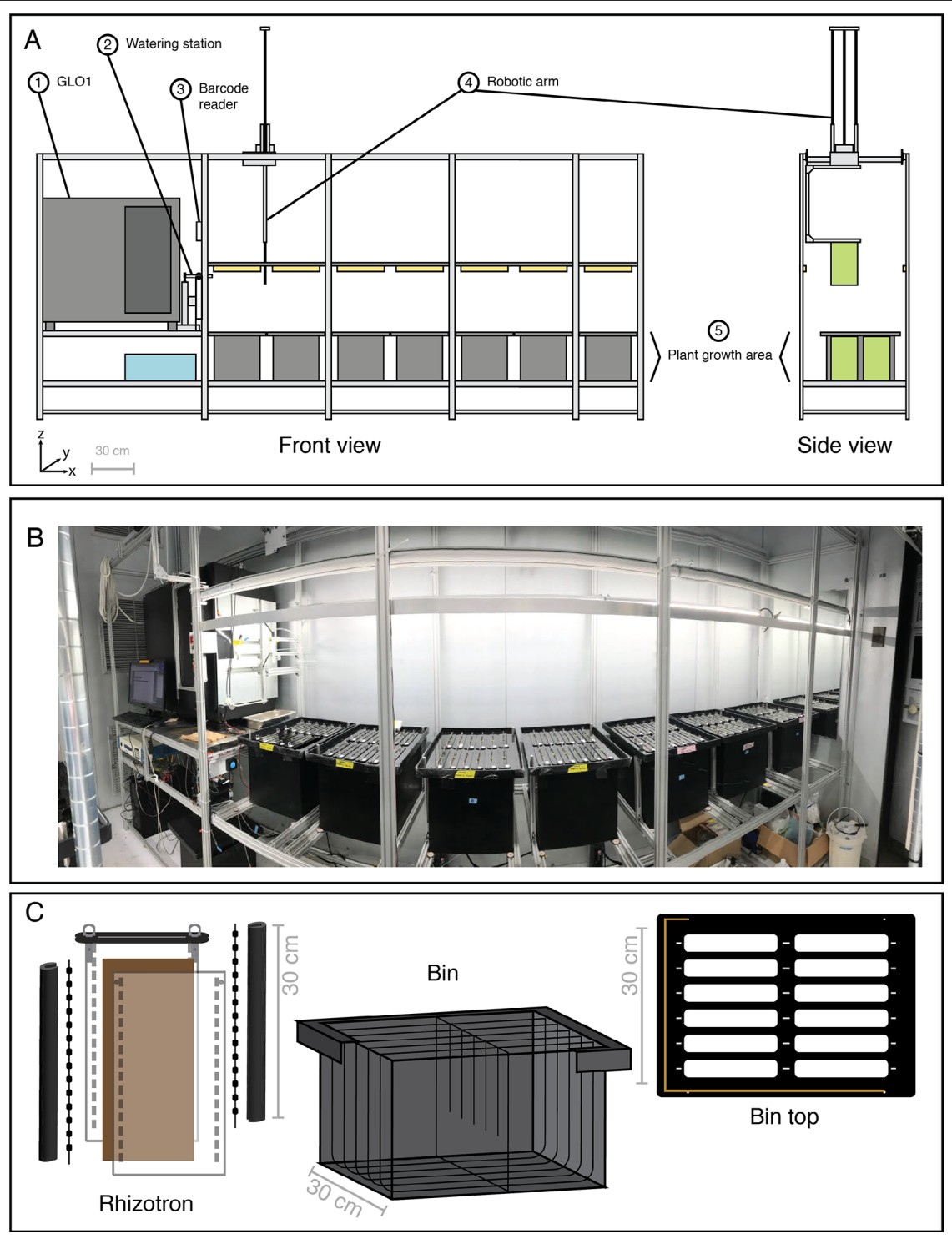

**Figure 1.** Growth and Luminescence Observatory for Roots (GLO-Roots) automation: GLO-Bot. (**A**) Schematic of the GLO-Bot Cartesian gantry system includes: (1) The Growth and Luminescence Observatory 1 (GLO1) imaging system published in *Rellán-Álvarez et al., 2015*, which houses two cameras and a rotating stage for root imaging, (2) a station for general watering or treatment with diluted luciferin solution prior to imaging, (3) a barcode scanner to identify the rhizotron and load a specific watering and imaging protocol, (4) a robotic arm, which moves in the x-, y-, and z-directions and has a hook at the end to pick up rhizotrons, and (5) an area for plant growth, which can be seen in the photograph of GLO-Bot (**B**). (**C**) Automation updates required modification of the GLO-Roots growth vessel design to include a black acrylic plate and hooks for rhizotron handling as well as dividers within the growth boxes and guides along the bin top, which allow the rhizotron to hang and shield the roots from light. Copper tape along the edge of the bin top enables positioning. Gray-scale bars denote 30 cm.

*Figure 1 continued on next page*

*Figure 1 continued*

The online version of this article includes the following figure supplement(s) for figure 1:

**Figure supplement 1.** Detailed schematic of rhizotron design for automation.

by sequentially adding images together. However, in addition to summing the root signal, this step also sums background noise (*Figures 2 and 3*). Therefore, we utilized a combination of pre- and post-measurement processing to amplify true signal and reduce noise (*Figure 2*). To account for some of the inherent noise introduced by the imaging system itself, we run the background removal macro remove_background.ijm, to subtract a blank image, taken in GLO1 without a rhizotron present, from every root image (*Figure 2A-2*). True root signal can be amplified, or inferred from previous days in the time-lapse, if all images of a root system have the root in the exact same position relative to the exterior image dimensions. Therefore, each series of root images is manually checked to ensure root position remains stationary over time. Misalignments were fixed using the ImageJ plugin Template Matching and Slice Alignment (Template_Matching.jar, *Tseng et al., 2011*). This plugin works by user selection of a region present in all images followed by automated detection and translation (x-, y-movement) of subsequent images to align image features, a method that is also known as image registration. This step occasionally required subsetting the images to yield precise alignment. During the alignment process, all root images were screened and roots were checked for: (1) appropriate number of images; (2) erroneous image dates; (3) excessive background noise; (4) any other anomalies from image testing (described in the Materials and methods 'Image analysis' section) or occasional camera malfunction. An incorrect number of images or erroneous image dates were due to technical issues with the automated plant handling and imaging pipeline. Images with excessive background noise, or extraneous images, were removed. Next, aligned and cleaned-up images were run through the macro denoise_overlay.ijm, designed to strengthen signal intensity and decrease noise through a series of image-wide addition and multiplication steps (see detailed steps in the Materials and methods 'Image analysis' section). The output from this macro is a set of images in which each day of imaging is added to the previous days, thus accumulating the strongest signal throughout the root system (*Figures 2A–4*).

With luminescent root signal maximized throughout all of the images in a time-lapse series, the macro invert.ijm is used to invert the gray-scale values of the images, thus making the root system itself black and the background white for downstream whole root system analyses. In addition to determining architectural parameters of complete root systems, we also established methods to quantify the actively growing regions of root systems. For this, we use the macro tip_tracking.ijm to isolate the new growth between images by a series of dilating and subtracting of two consecutive images (see detailed steps in the Materials and methods 'Image analysis' section; *Figures 2A–5*).

## Downstream analyses of root growth dynamics requires the distinction between true root signal and background pixels

All images were run through a modified version of GLO-RIA (Growth and Luminescence Observatory Root Image Analysis; *Rellán-Álvarez et al., 2015*), GLO-RIAv2 (*Figures 2A–6*). GLO-RIAv2 removes features deprecated from the 2017 ImageJ update (*Schneider et al., 2012*; *Schindelin et al., 2012*) and includes root angle output ranging from 0° to 180°.

Downstream analyses of the GLO-RIAv2 outputs were performed in R (*R Development Core Team, 2019*). Most root system structures visible on the GLO-Bot images are disconnected in many locations, due to variability in the reporter's intensity and the presence of physical obstructions, such as soil particles. We call each detected bioluminescent piece of root a 'root segment'. Data for each root segment include (1) the x-, y-coordinate of the upper left corner of the bounding box for each segment, (2) the segment length, and (3) the angle with respect to gravity,

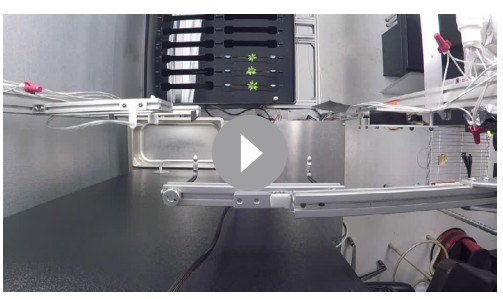

**Video 1.** GLO-Bot running.
https://elifesciences.org/articles/76968/figures#video1

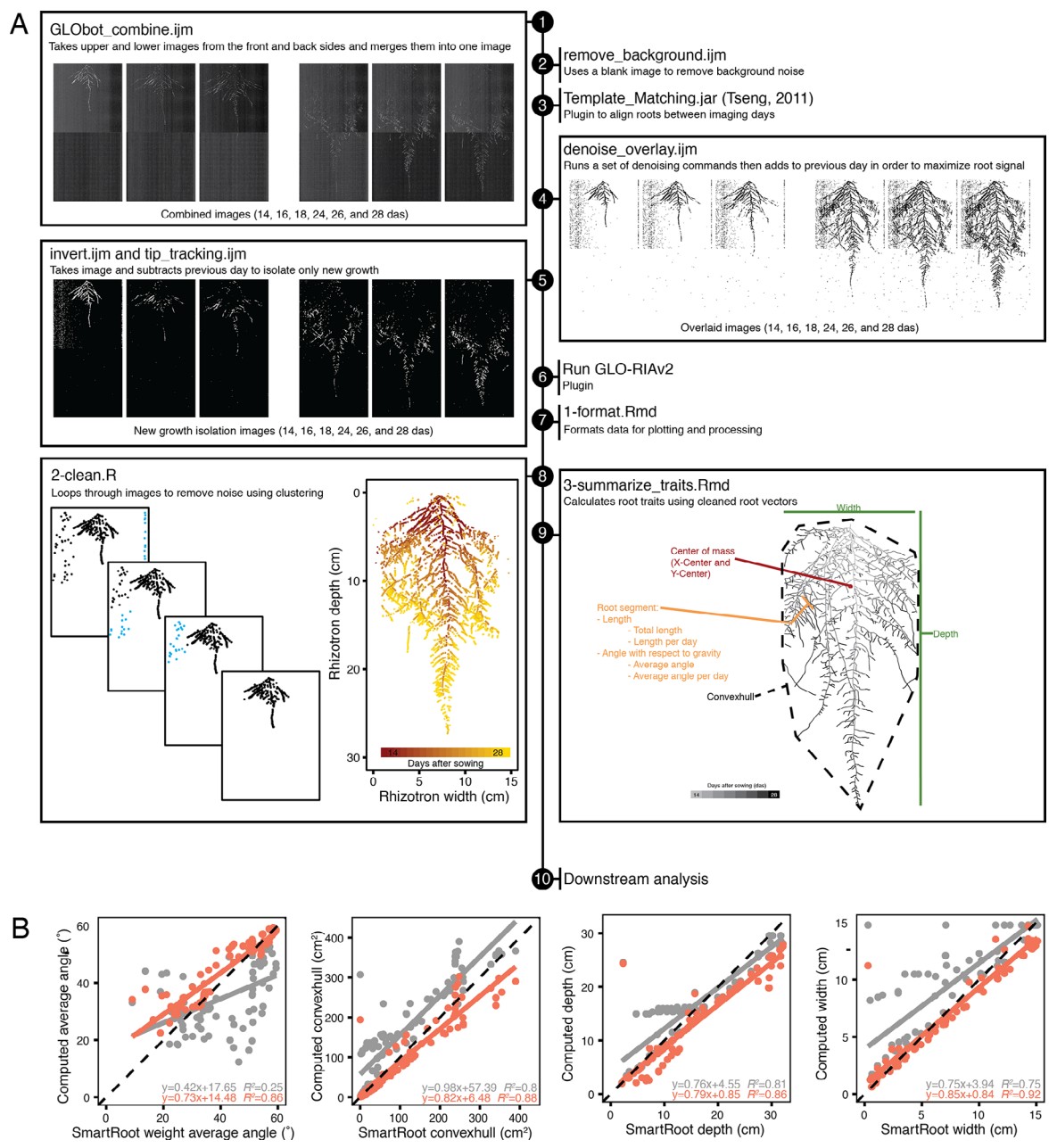

**Figure 2.** Workflow of image analysis pipeline that enables robust trait measurement. (**A**) Sample workflow for processing time-series images from multiple days after sowing (das) starts by (1) combining raw images to merge front, back, upper, and lower images, (2) a blank image is then subtracted to remove inherent noise, (3) images are registered using the ImageJ plugin Template Matching and Slice Alignment to account for x- and y-movement caused by slight position changes during rhizotron insertion into the imager, (4) registered images are then de-noised and overlaid, which helps overcome luciferase signal loss in older root tissues; however, this step also compounds any background noise. (5) Image subtraction between each day removes this noise and isolates new growth. Processed images are run through GLO-RIAv2 (6) and the output is formatted, such that in silico vectorized roots can be reconstructed (7). Additional noise is cleaned up (8) using an iterative distance-clustering method. Traits are then extracted from cleaned roots (9) and the outputs can be used for downstream analysis (10). (**B**) Comparison of 85 images measured with previous root analysis methods (GLO-RIAv1, gray) and the new analysis methods (GLO-RIAv2, coral) vs. manually traced ground truth measurements from SmartRoot (x-axis) reveals that the new analysis method increases accuracy, as demonstrated by the $r^2$ values, higher slopes, and lower y-intercepts.

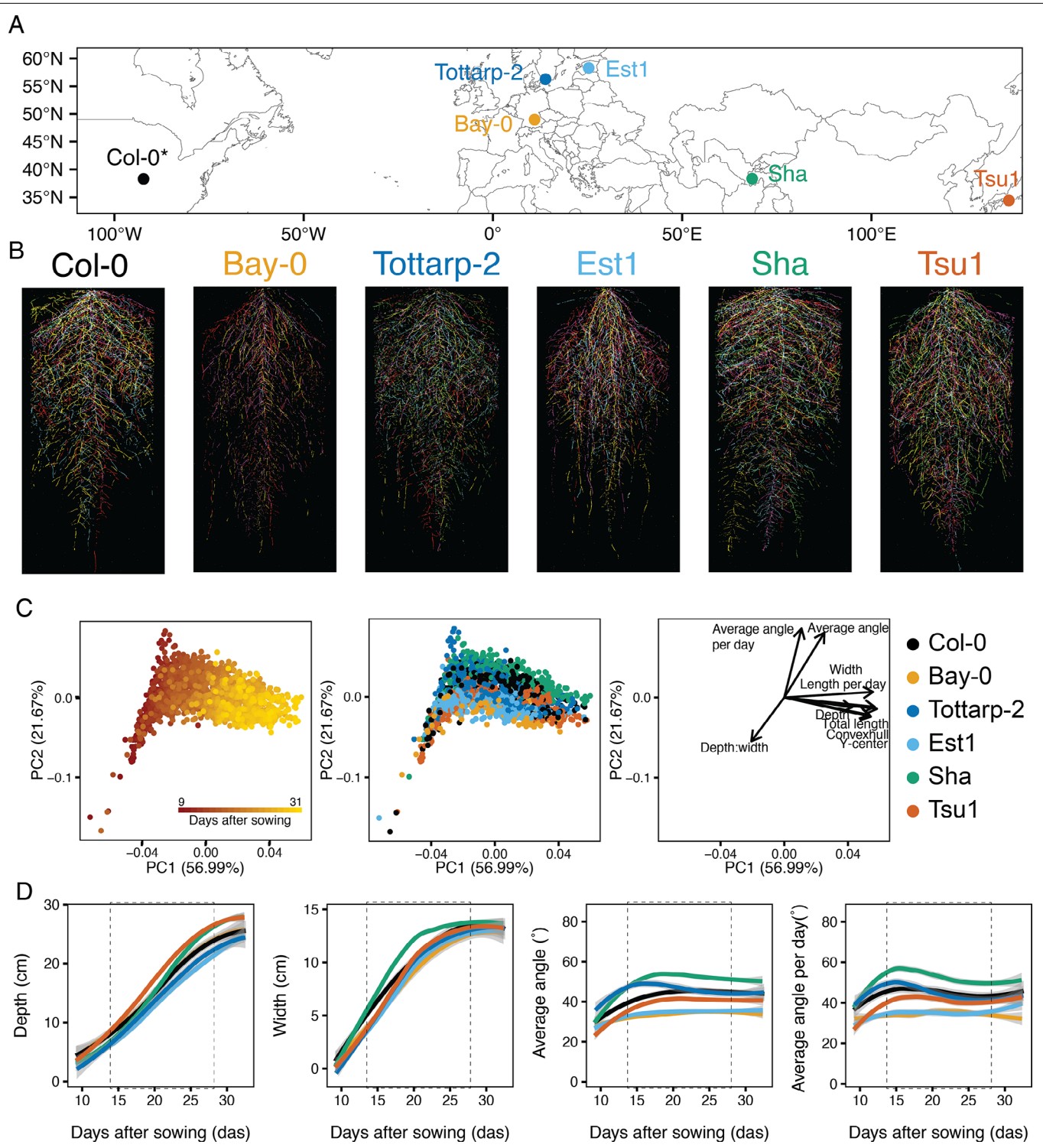

**Figure 3.** Root system architecture (RSA) of six *Arabidopsis* accessions over time. (**A**) Six accessions from diverse locations were imaged continuously from 9 to 31 days after sowing (DAS). *Col-0 location reflects where accession was originated, not collected. (**B**) Each accession had a unique RSA pattern (five root systems per accession at 21 DAS each shown in a different color overlaid on top of each other). (**C**) Principal component analysis (PCA) using raw trait values for eight time points and nine traits shows time (red is 9 DAS and yellow is 31 DAS), as the largest source of variation (PC1) and angle and depth-to-width ratios further distinguishing the root systems (PC2). When colored by accession, this analysis shows separation between the accessions. Loading plot illustrates the impact of each trait on the overall variation. (**D**) Four root system raw traits for six accessions growing over time from 9 to 31 DAS. Trends were calculated using LOESS smoothing (n=12–15 plants at each time point). 95% confidence interval shown by gray shading.

*Figure 3 continued on next page*

*Figure 3 continued*

Non-smoothed raw data and additional traits are plotted in *Figure 3—figure supplement 2*. Two vertical dashed lines highlight the interval (14–28 DAS) chosen for future experiments. Depth and width measurements are constrained by the rhizotron size. Colors correspond to accessions in panel B.

The online version of this article includes the following figure supplement(s) for figure 3:

**Figure supplement 1.** Principal component analysis (PCA) plots of nine root system architecture (RSA) traits for the six accessions.
PCA using raw trait values for nine traits. Left plots show PCA colored by time (red is 9 DAS and yellow is 31 DAS), center shows PCA colored by accessions, and the right shows the loading plot to demonstrate the traits influencing each PC.

**Figure supplement 2.** Root system architecture (RSA) traits over time for the six accessions.

**Figure supplement 3.** Continuous vs. single time point imaging.

such that root segments can be referenced with 0° pointing straight down and 90° pointing straight out to the side. These data give us the ability to reconstruct the root system in silico as a series of vectors (*Figure 2*).

Next, we remove non-root particles by iteratively clustering the starting points of the measured segments. Clusters are tested for distance between points, the number of segments, and the linearity of the points within the cluster (detailed steps in the Materials and methods 'Image analysis' section, *Figure 2A-8*). Segments that do not meet empirically defined criteria are removed, leaving behind only 'true root' segments. Using the 'true root', we calculate traits associated with new growth including: number, length, and angle of new root segments, as well as traits at the whole root system level: width, depth, center of mass, convexhull, depth-to-width ratio, total length, and average angle (*Figure 2A-9*). When computing average angle measurements, vector segments are weighted by length, meaning that small segments contribute less to the overall trait value compared to longer root segments. In addition to these seven traits, new root length (length per day) and average angle (average angle per day) are also summarized on a per-day basis. Additional traits can be derived from these metrics and extracted from the raw images for future analysis. It is important to note that each computed trait must be considered in the context of the experimental system. For example: (1) width and depth measurements are ultimately constrained by rhizotron size making these traits relevant up to a certain time point depending on the accession, (2) correspondingly, the depth-to-width ratio will always converge close to 2:1 as the root system fills the rhizotron, and (3) root length is an approximation since it is a sum of the visible root segment lengths and therefore does not account for the distance between disconnected segments. Despite these caveats, the nine traits measured (listed above and summarized in *Supplementary file 1*) encompass the basic parameters commonly used to describe the root system as well as new traits that are made possible through our time-lapse imaging and automated plant handling system.

To test the accuracy of our trait extraction methods, we used SmartRoot (*Lobet et al., 2011*) to manually trace the root systems for five growing plants and ground truth our parameters. We observed a strong correlation between the traits extracted via our new image processing pipeline and those that were measured manually (*Figure 2B*). Manual SmartRoot tracing of a single GLO-Roots image can take multiple days, thus making analysis of large-scale time course experiments unfeasible. While the initial GLO-Roots pipeline works well for end-point image analysis, the above-mentioned updates allow for more accurate root trait measurements of time course images (*Figure 2B*) and gives us ways to explore new temporal traits.

## Time-lapse imaging of six *Arabidopsis* accessions reveals distinct root growth trajectories

The robotics and image analysis pipeline we developed allowed us to increase the throughput of the GLO-Roots system, which we then applied to address two important questions: How do root system traits vary over developmental time, and how do dynamic root system traits vary across a species? To address the first question, we wanted to characterize root system traits with high temporal resolution and with greater replication than possible with a larger survey of genotypes. We selected a set of six *Arabidopsis* accessions based on their geographic dissimilarity and availability of homozygous LUC2-expressing lines with strong reporter expression (*Figure 3A*; *Rellán-Álvarez et al., 2015*). Each accession was grown in 10 replicates, where 5 replicates were imaged daily from 9 days after sowing (DAS) until 31 DAS and the other 5 replicates from 21 to 31 DAS. In depth comparisons between replicates

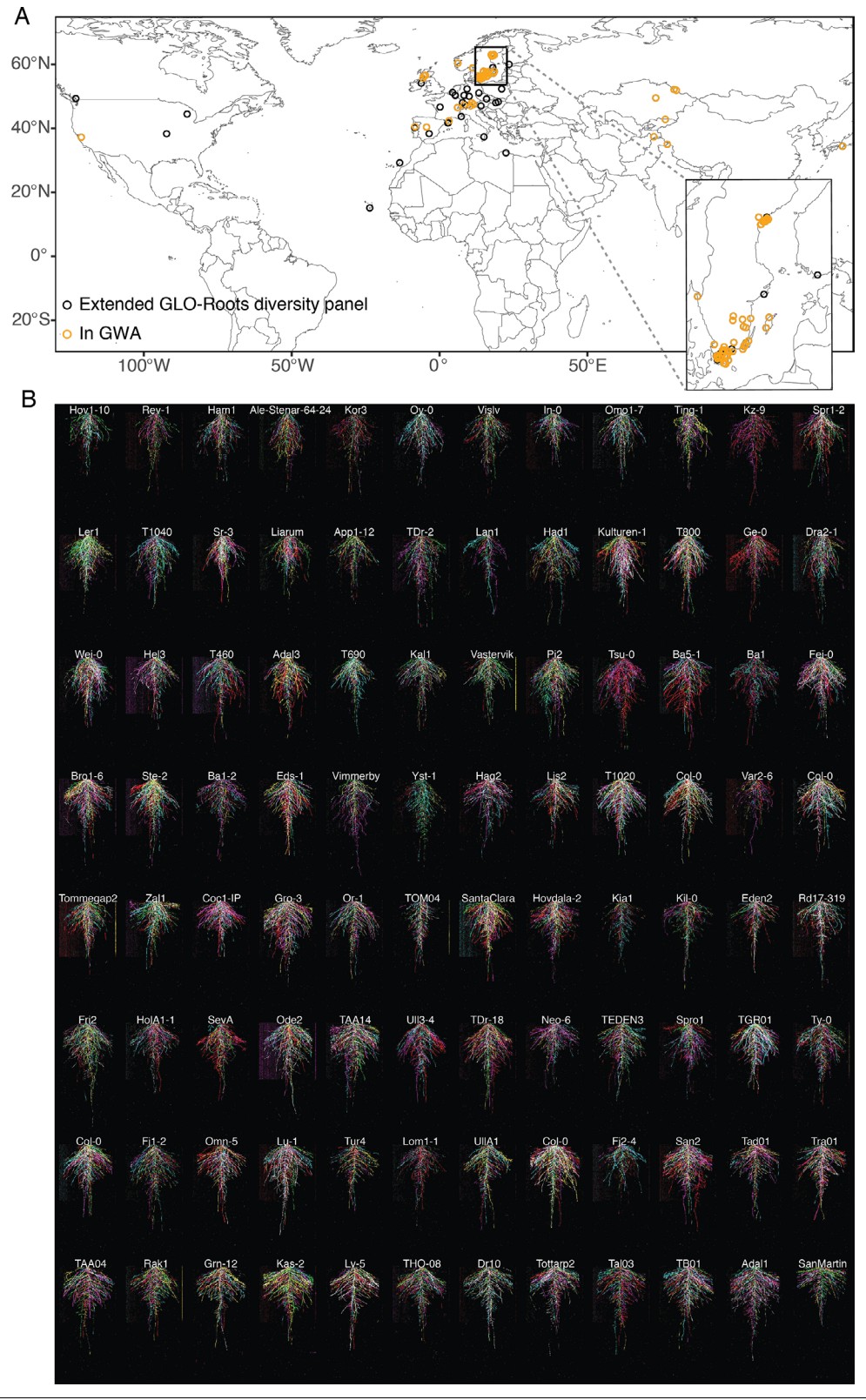

**Figure 4.** Root system architecture (RSA) of the Growth and Luminescence Observatory for Roots (GLO-Roots) diversity panel. (**A**) Locations of all accessions used in this study. Inset highlights the concentration of accessions from the Swedish population. (**B**) RSAs of the GLO-Roots diversity panel used in the genome-wide association studies (GWAS) at 20 days after sowing (DAS). Six root systems overlaid on top of each other with root system

*Figure 4 continued on next page*

*Figure 4 continued*

color indicating each replicate. Root systems are arranged in order of median average angle of the root system from deepest in the upper left to shallowest in the bottom right.

The online version of this article includes the following figure supplement(s) for figure 4:

**Figure supplement 1.** Overview of the Growth and Luminescence Observatory for Roots (GLO-Roots) diversity panel.

**Figure supplement 2.** Growth and Luminescence Observatory for Roots (GLO-Roots) diversity panel rhizotron positioning.

**Figure supplement 3.** Raw vs. fitted values.

---

can be done using the RShiny App (https://tslarue.shinyapps.io/rsa-6access-app/). The six accessions that were imaged displayed diverse RSAs (*Figure 3B*, *Video 2*) and principal component analysis (PCA) demonstrated that we could distinguish the accessions based on nine of the traits extracted using our image analysis pipeline (*Figure 3C*, *Supplementary file 1*). Expectedly, time accounted for the largest source of variation in PC1 (56.99%, *Figure 3C*), indicating that the size and developmental stage of a plant is the most important variable affecting RSA. We could see that depth-to-width ratios and average angle measurements further separated the accessions along PC2 (21.67%, *Figure 3C*). In total, the first four PCs capture 94.02% of the variation with the greatest distinction between accessions apparent in PC2 (*Figure 3—figure supplement 1*). Looking at the traits over time demonstrated that each of the accessions had its own unique growth trajectory with average angle and average angle per day exhibiting the most diversity (*Figure 3D*, *Figure 3—figure supplement 2*). Early during plant growth all accessions have fairly similar average angles, presumably because they have few lateral roots, but over time their average angles for the whole root system shift and distinguish the accessions. By looking at these data, we established that most traits stabilized by the end of the 28 DAS growth period (*Figure 3D*, *Figure 3—figure supplement 2*). Importantly, having imaged the plants both continuously and only at 21 DAS, we were able to confirm that these traits were not influenced by continuous imaging and the associated luciferin watering (*Figure 3—figure supplement 3*). Time-lapse imaging of these six accessions proved the robustness of GLO-Bot running for 15 hr a day over several weeks, the ability of our image analysis pipeline to determine various root system traits over time, and that even a small subset of accessions showed substantial variation in RSA traits. In addition, the application of our newly developed imaging protocols to these data allowed us to optimize experimental parameters, including luciferin watering speed and frequency, imaging frequency, and imaging time period. Those parameters were foundational to examine a larger, more variable population.

## Natural variation in dynamic RSA traits is highly complex and polygenic
### The GLO-Roots diversity panel
To explore the diversity of root architectures in the *Arabidopsis* species and define the genetic basis of these traits, we sought to apply the GLO-Roots method to a larger panel of curated accessions. We therefore transformed an initial set of 192 *A. thaliana* accessions with the *ProUBQ10:LUC2o* reporter for use with the GLO-Roots system (*Rellán-Álvarez et al., 2015*; *Figure 4—figure supplement 1A, B*). These accessions comprise the extended GLO-Roots diversity panel, which originated from a densely sampled Swedish population (*Long et al., 2013*) and were supplemented with accessions from extreme environments with high and low annual average precipitation (*Fick and Hijmans, 2017*), as well as accessions with high and low sodium accumulation levels in the shoot (*Baxter et al., 2010*). Ultimately, the GLO-Roots diversity panel, a set of 93 accessions, were selected for the GWA population (*Figure 4ASupplementary file 2*). This set balances genetic similarity and environmental variation by combining 72 Swedish accessions with 20 accessions from extreme environments. These lines were selected based on location, reporter expression strength, and the number of independent transformation events recovered to avoid insertion effects. Four Col-0 control plants grew in different growth boxes to examine position effects (*Figure 4—figure supplement 2*). This population not only represents a foundational resource for describing phenotypic trait diversity in the *Arabidopsis* species, but can later be used to identify genes underlying this trait diversity over time, using the tool of GWA mapping.

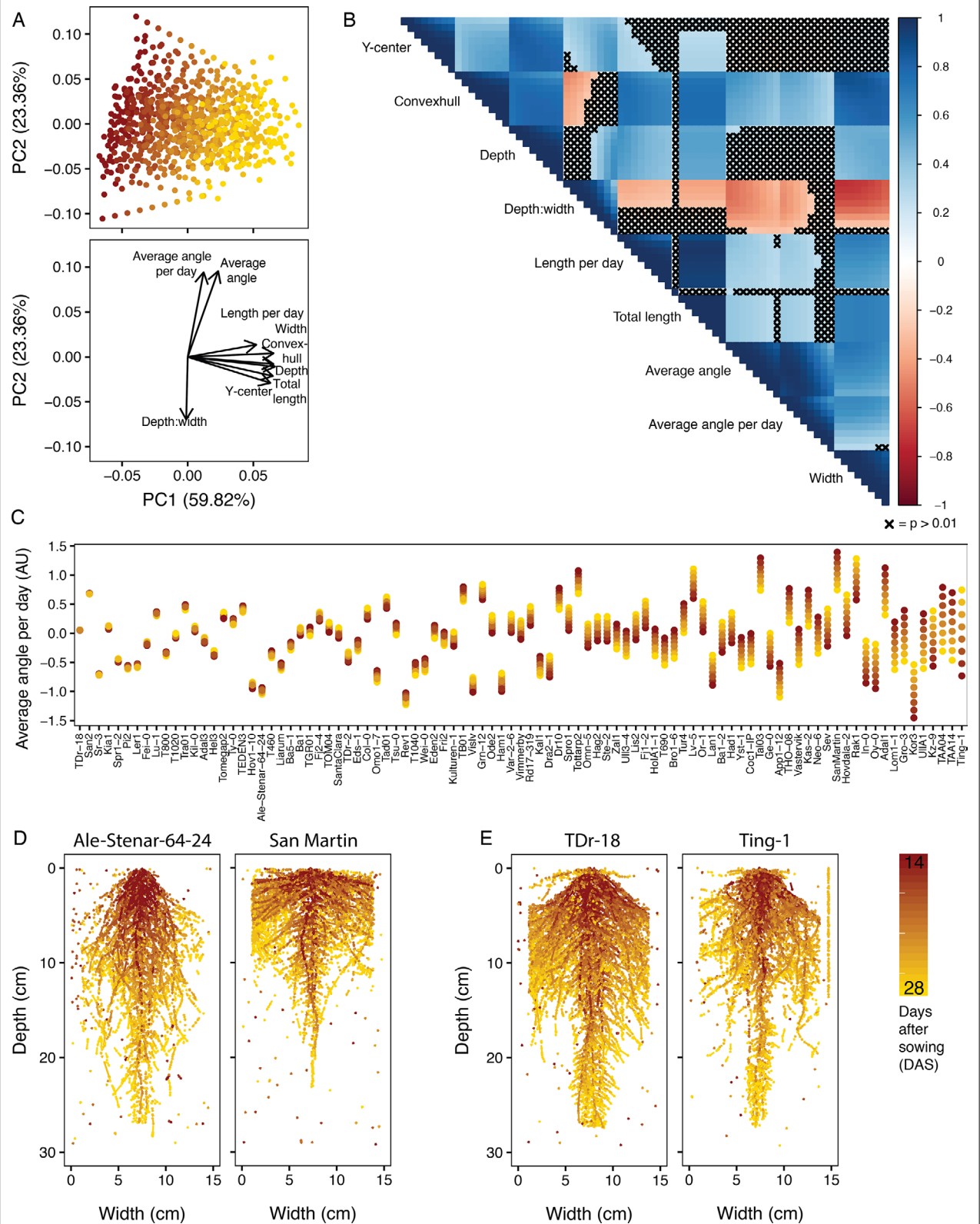

**Figure 5.** Trait relationships among the Growth and Luminescence Observatory for Roots (GLO-Roots) diversity panel. (**A**) Principal component analysis (PCA) of nine fitted traits (i.e. fitted values) for the 93 accessions of the GLO-Roots diversity panel reveals traits influenced by time as the predominant source of variation within the dataset (PC1), while angle measurements and the depth-to-width ratio further distinguish accessions (PC2). (**B**) A correlogram of the fitted trait values at each time point, increasing left to right and top to bottom, further shows the negative correlations between the

*Figure 5 continued on next page*

*Figure 5 continued*

depth-to-width ratio and the angle traits and displays the relationships between all nine traits at each time point. Correlations where p>0.01 have been crossed out with an 'X'. While some traits remain stable over time, other traits change over time. The extent to which a trait changes varies between accessions. (C) In some accessions, the fitted angle of new growth (average angle per day) remains constant each day, while in others the angle of new growth increases or decreases over time. The accessions are ordered based on variability of average angle per day with similar angles to the left and strikingly different angles on the right. (D) Ale-Stenar-64–24 and San Martin demonstrate contrasting root architectures, with opposite depth-to-width ratios and opposite average angles. (E) TDr-18 and Ting-1, on the other hand, are examples of accessions with constant and variable (respectively) average angle per day measurements.

The online version of this article includes the following figure supplement(s) for figure 5:

**Figure supplement 1.** Principal component analysis (PCA) for nine root traits describing root system architecture (RSA).

Although GLO-Bot could image 96 rhizotrons in 24 hr, we decided to run it at half capacity in order to have time to respond and adjust to technical errors as they arise. To capture the breadth of RSA dynamics within the diversity panel, we imaged 48 rhizotrons on 1 day (*Figure 4—figure supplement 2*, part 1) and the other 48 rhizotrons on the next day (*Figure 4—figure supplement 2*, part 2) from 14 to 28 DAS (*Figure 4B*, *Video 3*, *Figure 4—figure supplement 1C*). We grew all 93 accessions at once and replicated the experiment six times. Next, using our image analysis pipeline, we focused on the nine root traits that distinctly separated RSAs in the subset of six accessions, and which encompass RSA at a broad scale (*Supplementary file 1*).

Growth of root systems in soil is likely to incur an experimental cost due to the greater variability in experimental conditions, compared to gel-based media, and the resultant increase in variability of the root traits measured. Furthermore, the size of the diversity panel we used required that biological replication be performed on separate dates, due to limits in the capacity of the automation pipeline. To account for these additional sources of variation, we developed an analysis pipeline that allowed us to summarize trait data for our replicated time course experiments. For each measured trait, we estimated 'fitted values' to account for replicate noise (*Figure 4—figure supplement 3*). Fitted values were calculated using a generalized linear mixed model for each trait using the R package MCMC-glmm (additional information can be found in the Materials and methods 'Trait processing' section; *Wilson et al., 2010*; *Mrode, 2014*; *Hadfield, 2010*). The model provided an intercept and slope for each genotype which were used to calculate the expected values at 0, 48, 96, 144, 192, 240, 288, and 336 hr after the start of imaging (equivalent to 14, 16, 18, 20, 22, 24, and 28 DAS). These fitted values were used for further analyses.

## Root trait diversity and phenotypic relationships among the GLO-Roots diversity panel

Similar to the analysis of the first set of six accessions (*Figure 3C*), PCA of the GLO-Roots diversity panel revealed that the first principal component captures trait variation over time (*Figure 5A*, *Figure 5—figure supplement 1A*), as well as traits that are strongly associated with time and, thus, developmental stage and plant size (*Figure 5A*, *Figure 5—figure supplement 1B*) such as the root system's area (convexhull area), length (total length), and depth. The second principal component demonstrates that both the average angle of the root system and the average angle of new growth account for a large amount of root system variation (*Figure 5A*, *Figure 5—figure supplement 1C*, *D*). Average root angle and depth-to-width ratio are negatively correlated (*Figure 5A, B*), as demonstrated by the accession San-Martin, which has the smallest depth-to-width ratio and the highest average root angle (i.e. most shallow) as well as Ale-Stenar-64-24, which has one of the highest depth-to-width ratios and lowest average root angles (*Figure 5D*). Additional data exploration and accession comparisons can be done using the RShiny App (https://tslarue.shinyapps.io/rsa-app/), which plots the vectorized root systems for all of the accessions. The raw data is available through Zenodo at https://doi.org/10.5281/zenodo.5709009. Accessions and replicate numbers with excessive background noise or other anomalies are identified in *Supplementary file 3*.

Further analysis of the traits via pairwise comparison at all time points not only allowed us to visualize existing correlations within traits over time, but also among traits (*Figure 5B*). Looking at average angle per day, we see that consecutive days show strong correlation across accessions (e.g. day 14 compared to day 16: Spearman's rank correlation 0.99), whereas later time points are less well correlated with early time points (day 14 compared to day 28: Spearman's rank correlation 0.56,

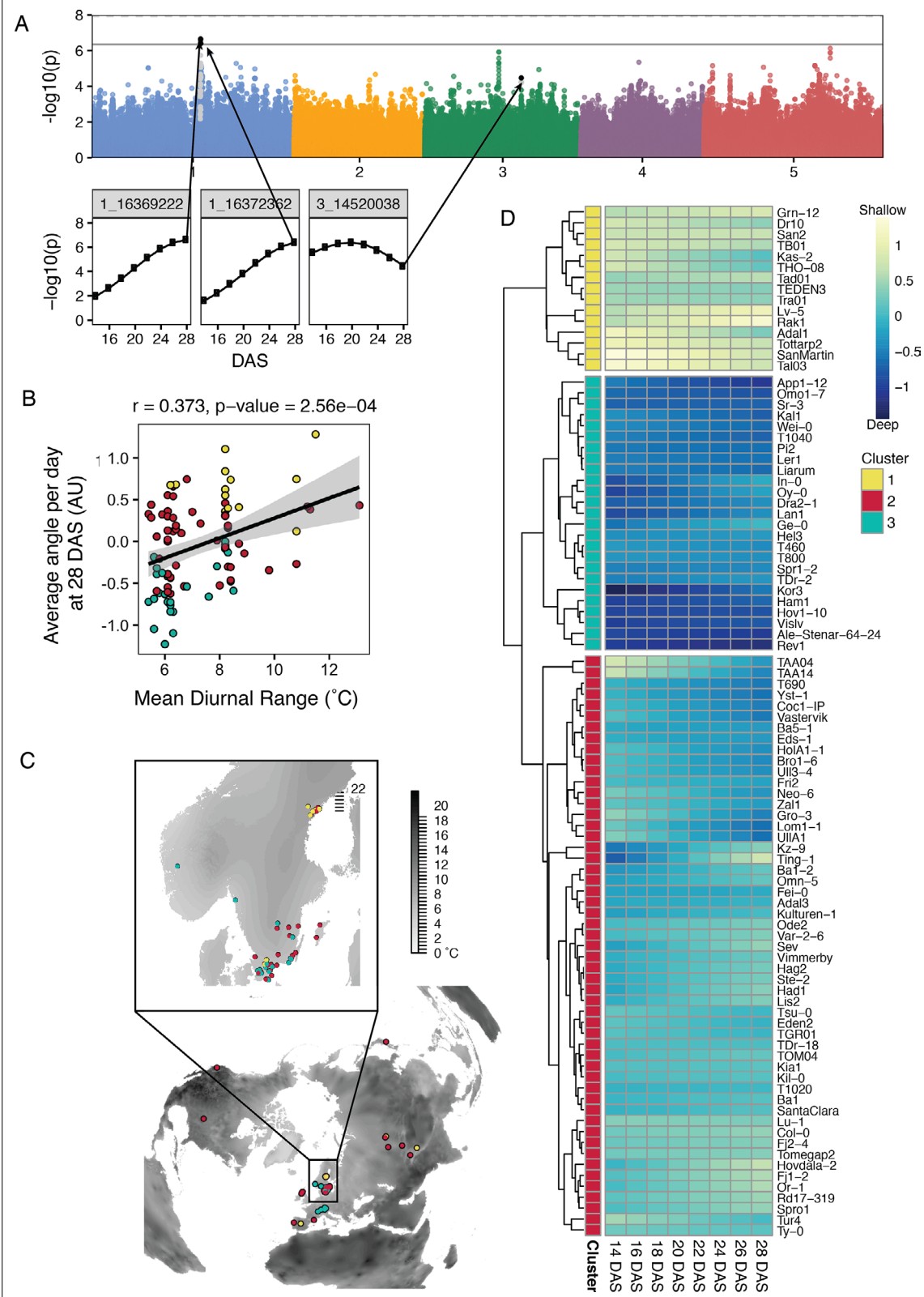

**Figure 6.** Genome-wide association studies (GWAS) and climate correlations of root traits of the Growth and Luminescence Observatory for Roots (GLO-Roots) diversity panel. (**A**) Manhattan plot for average angle per day at 28 days after sowing (DAS). Black points indicate SNP positions that pass the Bonferroni threshold (solid line) at least once throughout the time series, as shown in the inset plots. Gray points are those in linkage disequilibrium (LD) with the significant SNPs. (**B**) Correlation between the average angle per day and the mean diurnal temperature range indicates the potential

*Figure 6 continued on next page*

*Figure 6 continued*

importance of root angle for surviving highly variable climates. (**C**) A world map showing mean diurnal range (black represents larger fluctuations) with the distribution of accessions (points). Points in (**B**) and (**C**) colored by cluster identity based on changes in average angle per day over time and computed by between-group average linkage hierarchical clustering with distance = 1.75, as shown in the heatmap (**D**). Heatmap depicts average angle per day for each day with blue indicating steeper angles and yellow indicating shallower angles. Changes in average angle per day divided the accessions into three clusters: the yellow cluster with consistently shallow root growth, the blue cluster with consistently deep root growth, and the red cluster with intermediate or changing root angles.

The online version of this article includes the following source data and figure supplement(s) for figure 6:

**Source data 1.** Manhattan plots and SNP development of all nine root system architecture (RSA) traits.

**Figure supplement 1.** Correlations between average climatic variables and root traits.

*Figure 5B*). This suggests that the average angle of lateral root tips is a trait that changes over time with some accessions showing greater change than others (*Figure 5C*). Indeed, examining this trait across our 93 accessions reveals great diversity in how stable lateral root tip angle is across development (*Figure 5C*). While many accessions have a consistent angle of new growth, such as TDr-18 and San2 (*Figure 5C*, *Figure 5E*), others change their angle of new growth over time, such as TAA14 and Ting-1 (*Figure 5C*, *Figure 5E*).

## RSA traits are heritable and highly polygenic

We next wanted to investigate the genetic basis of RSA traits. Some traits, such as the angle measurements, had high broad-sense heritability while others, such as total length, had low broad-sense heritability (*Supplementary file 5*). While some of the low heritabilities could be due to the limitations of the experimental method of growth or trait measurement, these data indicate measurable genetic control over some traits, which led us to use GWA to identify potential causal loci. Using the fitted values, we conducted GWA analyses for all nine traits at each time point using the AraGWAS database SNP matrix and the program GEMMA (*Zhou and Stephens, 2012*; *Togninalli et al., 2018*). In total, 29 SNPs were significant for at least one time point using Bonferroni multiple-test p-value correction (*Supplementary file 4*). These alleles also had a minimum allele frequency greater than 0.05.

These SNPs highlight regions of the genome with elevated significance and the temporal nature of our data allows us to track the changes in significance over time (*Figure 6A*, *Figure 6—source data 1*). For example, when looking at average angle over time, we see a greater strength of association, as indicated by lower p-values over time, which likely corresponds to the stabilization of the trait (*Figure 6A*, *Figure 6—source data 1B*). In contrast, we see the strength of the association decrease for depth-to-width ratios over time (*Figure 6—source data 1B*), which likely reflects that growth of root systems becomes constrained at later time points by the physical limits of the rhizotron. We gain confidence in the chromosomal regions identified by tracking the

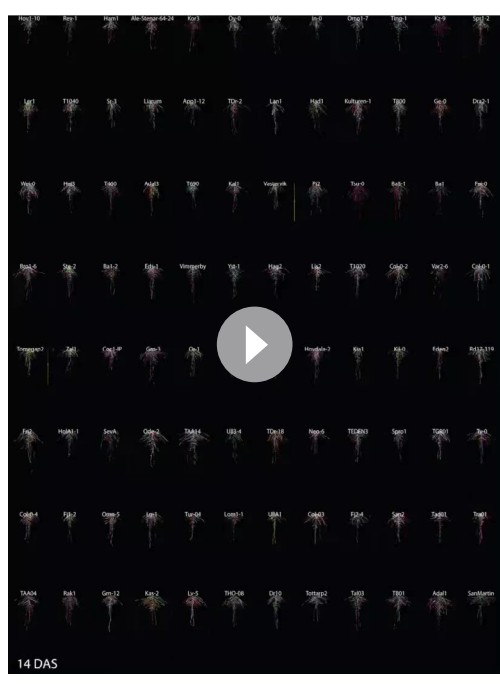

**Video 3.** Root system architecture (RSA) growth of Growth and Luminescence Observatory for Roots (GLO-Roots) diversity panel over time.
https://elifesciences.org/articles/76968/figures#video3

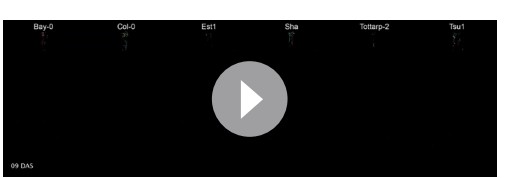

**Video 2.** Root system architecture (RSA) growth of six *Arabidopsis* accessions over time.
https://elifesciences.org/articles/76968/figures#video2

significance over time and observing the slight enrichment of experimental p-values in our Q-Q plot distributions (*Figure 6—source data 1*).

Out of the 29 identified significant SNPs, four synonymous SNPs can be found within the coding sequence of two genes, eight were in intergenic regions, whereas the majority of SNPs were found in upstream- and downstream regions of genes, potentially affecting expression patterns (*Supplementary file 4*). It should be noted that the most significant SNP is not always the causal SNP. For this reason, *Supplementary file 4* lists all of the genes within a 20 kb region surrounding the SNP (10 kb upstream and 10 kb downstream). The identification of many significant SNP associations suggests that RSA development is a polygenic trait.

Looking at the two traits with the highest heritability (*Supplementary file 5*), average angle ($H^2$=0.45) and average angle per day ($H^2$=0.31), we found 12 and 8 significant SNPs, respectively (*Supplementary file 4*). Interestingly, both traits revealed one common SNP upstream of a hypothetical protein (AT3G42390) with 30% protein identity to a phospholipase D alpha 3 (At5g25370) (*Supplementary file 4*). Furthermore, the only intragenic SNP for average angle per day was found in an intron of another hypothetical protein with 35% identity to a non-specific phospholipase C2. Phospholipases are lipid-hydrolyzing enzymes that are known to play a role in signaling during plant development, stress responses, and responses to environmental cues (*Takáč et al., 2019*). We identified SNPs in two genes encoding proteins that show similarity to phospholipases suggesting a potential role of phospholipids in adjusting root angles to local environments.

Root depth was associated with a SNP upstream of a gene encoding Cyclin A2;3 (AT1G15570; *Supplementary file 4*). This gene was previously found to play a role in auxin-dependent mitotic-to-endocycle transition that is involved in the transition from cell proliferation to cell differentiation in the *Arabidopsis* root meristem (*Ishida et al., 2010*). Thus, alterations in gene expression could influence root depth in certain *Arabidopsis* accessions.

The GWAS on the depth-to-width ratio yielded an intronic SNP in a P-loop nucleoside triphosphate hydrolases superfamily protein with Calponin Homology domain-containing protein (*Supplementary file 4*). This gene is expressed in the lateral root cap (http://bar.utoronto.ca/eplant/; *Waese et al., 2017*) and changes in expression may influence root gravitropism leading to a more shallow root system. Future work will establish the molecular basis for the SNP-phenotype relationships identified here.

## RSA traits of the GLO-Roots diversity panel significantly correlate with climatic variables

To gain insight into the relevance of the different root architectures of the GLO-Roots diversity panel in the natural environment, we correlated our nine root traits with published bioclimatic variables (*Fick and Hijmans, 2017*). We see the strongest correlations between these traits and climatic variables related to temperature (*Figure 6—figure supplement 1*). Specifically, average angle per day shows a notable correlation to mean diurnal temperature range (Pearson r=0.373, p=2.56 × 10$^{-4}$; *Figure 6B*). This result indicates that accessions with shallow RSAs tend to grow in climates that exhibit larger changes in the daily minimum and maximum temperatures. Likewise, convexhull shows a positive correlation to mean diurnal range, albeit less significant (*Figure 6—figure supplement 1*). These data suggest that plants with shallow roots may actually grow better than those with deep roots in a more variable environment. Clustering of accessions by change in average angle per day over time divides the accessions into three groups: consistently shallow, consistently deep, and intermediate or changing root angle (*Figure 6C, D*). Interestingly the consistently shallow and consistently deep accessions segregate from each other in a global map, which is consistent with our finding that climate is likely to be an important variable determining the distribution of accessions with these traits (*Figure 6C*).

## Discussion

Implementation of automation expands the number of individuals that can be phenotyped and the phenotypes that can be captured (*Gehan et al., 2017*). By automating the GLO-Roots system, we created GLO-Bot, a robotic platform that enables unprecedented insight into *Arabidopsis* root growth over time in soil and at developmental stages rarely observed. Careful design of the system to facilitate

automated handling allowed us to maintain physiological relevance while increasing throughput and allowing for novel trait measurements.

Along with the automated imaging system, we developed an improved image analysis pipeline, GLO-RIAv2, for quantification of RSA. The previous image analysis program was semi-automated, and often needed intervention to define regions of interest. While this is feasible for end-point imaging with a small number of samples, it was not scalable with GLO-Bot. For automated image analysis, however, one of the biggest challenges was the combination of maximizing root luminescence signal intensity while decreasing background noise so that GLO-RIAv2 would only analyze true root signal. We saw a strong correlation between our traits extracted via our new image processing pipeline and those that were measured manually, demonstrating that our updates allowed for accurate examination of growth dynamics in both new growth and at a whole root system level. Recent progress in machine learning is demonstrating the opportunities within the field for segmenting out root and noise and could further improve accuracy (*Wang et al., 2019*; *Smith et al., 2018*). Whereas RGB imaging together with software tools like RootPainter has facilitated automated root system analysis in species with thicker roots (*Smith et al., 2020*; *Bauer et al., 2022*), bioluminescence still has advantages with species with finer roots, like *Arabidopsis*, since it improves the contrast between the root and soil signal. Using transgenic lines homozygous for the luciferase transgene would lead to stronger signal and therefore better root detection. New advances in recapitulating a full bioluminescence pathway in planta (*Khakhar et al., 2020*) could provide an exciting alternative for increasing root luminescence signal and therefore simplifying image segmentation.

The automation of GLO-Roots allowed for imaging larger sample numbers, which enabled the visualization of root growth of different *Arabidopsis* accessions over time. At maximum capacity, GLO-Bot could image 96 rhizotrons within 24 hr. However, all experiments were run at approximately half capacity in order to facilitate the management of handling errors by the system. The biggest source of errors occurred during the return of a rhizotron back to its growth box, which we addressed by adapting the design of the holding bins. Further advances in robotics and in the precision with which components of the system are manufactured will likely improve the throughput further.

Our automated plant handling and image analysis pipeline allowed us to explore the importance of time and genetic diversity in driving changes in RSA. A set of six accessions that were homozygous for the luciferase reporter allowed us to image root systems every day with a high degree of replication. Such experiments also tested the robustness of the system and facilitated optimization of imaging parameters and watering times to extract root traits that reflect true differences in RSA. Nine root traits were automatically extracted through GLO-RIAv2 and showed striking differences even in this small subset of accessions. Our analysis of these data revealed that differences between accessions often only emerged over time, such as average root angle, which highlights the importance of quantifying root traits over time and at these later developmental stages.

The GLO-Roots diversity panel provided access to a greater amount of genetic variation, but required changes in experimental design and data analysis methods. The GWAS ultimately led to the identification of significant SNPs associated with variation in six of nine root traits measured. However, given the polygenic nature of these traits, our ability to detect individual causal SNPs or peaks may be low. Additionally, our relatively small population size (93 accessions) challenges our ability to discover SNPs significantly associated with the traits (*Gibson, 2012*). Although a surprisingly rapid linkage decay in *A. thaliana* (*Nordborg et al., 2005*) allows within-gene trait mapping (*Atwell et al., 2010*; *Exposito-Alonso et al., 2018*), it will be ideal to follow up the SNPs identified above and other close variants through further analyses, such as accession-specific gene expression and complementation, knock-out and reporter gene assays (*Ogura and Busch, 2015*). Besides the analyses conducted in this study, the generated datasets present valuable resources for further studies related to the genetic analysis of the principal components to explore the phenotype space, multi-trait analyses, modeling of phenotypic variation across time, and structure-function modeling between RSA and water use.

The correlation of our nine extracted root traits to published bioclimatic data from the origin sites of the respective accessions reveals a significant correlation between average angle to mean diurnal temperature range. Thus, accessions with shallow RSAs tend to grow in climates that exhibit larger changes in the daily minimum and maximum temperatures reminiscent of desert-like climates. This fits with the known pattern that many plant species growing in the desert show rather shallow root systems with advantages including reduced energy input, increased ability to capture moisture from

precipitation, and high nutrient availability in upper soil layers, which may improve survival in a more varied climate (*Pierret et al., 2016*; *Schenk, 2008*; *Ogura et al., 2019*).

Surprisingly, we found low or some non-significant correlations between the nine root traits and total annual precipitation, a factor we had initially used to select the accessions characterized in this study. This finding could be due to a variety of factors underlying the design of our study including the limited number of accessions analyzed, the specific selective pressures acting on Swedish *Arabidopsis* populations, where the majority of our characterized accessions originate (*Long et al., 2013*), and the well-watered conditions that we used to grow our plants. It may also be that annual precipitation is less important as a selective pressure acting on *Arabidopsis* root architecture compared to temperature. Indeed vapor pressure deficit (VPD), which is the driving force determining the rate of water loss during transpiration (*Fricke, 2017*), is exponentially related to temperature and modern crop plants appear to be particularly susceptible to elevated VPD (*Lobell et al., 2014*). To reliably link environmental parameters to RSA responses, it will be necessary to survey our accessions in conditions similar to their native environment or in conditions replicating environmental stresses. This will ultimately expand our understanding of how root architectural traits relate to different climate parameters and how the root system will react to a warming climate, both in agricultural and in natural ecosystems.

# Materials and methods
## Growth system
### Rhizotron design
As described in Rellán-Álvarez et al., 2015, rhizotrons were built with two sheets of 1/8" clear abrasion-resistant polycarbonate plastic (Port Plastics, Portland, OR, Product: MarkolonAR) and were water jet cut (AquaJet LLC, Salem, OR) into 15 cm × 30 cm rectangles with 14 small gaps running down each side (clear_sheets.dxf). Two spacers (spacers.eps) were laser cut (Stanford Product Realization Lab) from 1/8" black cast acrylic (TAP Plastics, Mountain View, CA, and Calsak Plastics, Hayward, CA). Two rubber U-channels were cut into 29 cm long strips from Neoprene Rubber Trim, 5/16" Wide × 23/32" High Inside (McMaster-Carr, Elmhurst, IL, #8507K33). Additional adaptations for automation included: beveling the top of each polycarbonate sheet to a thin 45° angle using a belt sander. Half of the polycarbonate sheets had 2.5 mm holes drilled in 5/8" down from the top edge and 3/16" in from each side. The small holes were used to screw a 1/16" aluminum water jet cut hook, outer dimensions 18 mm × 57 mm with a 13 mm diameter circle cutout (rhizotron-machine-hooks-13mm.pdf), to both sides of the sheet using 2×4 M3 stainless steel socket cap screws (McMaster-Carr, Elmhurst, IL, #91292A109). Not only did the hooks provide a way to pick up the rhizotron, they also were threaded through a rhizotron top (rhizo-bin-top-enlarged.pdf) cut (Pagoda Arts, San Francisco, CA) from 1/8" black acrylic, which served as a light shield and a bearing surface for the hanging rhizotron. The other half of the polycarbonate sheets had 8 mm holes drilled in centered at the same location and were used as counterparts to the sheets with tops and hooks.

### Boxes and holders
Similar to *Rellán-Álvarez et al., 2015*, rhizotrons were placed in black polyethylene boxes with acrylic holders during plant growth and imaging. Modifications were as follows: 12"W × 18"L × 12"H black polyethylene boxes (Plastic-Mart, Fort Worth, TX, Part number R121812A) were outfitted with a top holder and inner divider system to grow 12 rhizotrons at a time. The inside dividers were composed of 1/8" black acrylic cut into five sheets (bin-b.pdf, bin-c.pdf, and bin-d.pdf), which interlock perpendicularly with a large middle sheet (bin-d.pdf), as well as two smaller pieces (bin-b.pdf, bin-c.pdf) at each end. This divider system creates chambers, which isolate the rhizotrons from each other, thus preventing light affecting neighboring rhizotrons when one rhizotron is removed. Each bin has a bin top (rhiz-bin-top-enlarged.pdf), also cut from 1/8" black acrylic, has 12 cutouts arranged in two rows of six, where the rhizotrons sit. The cutouts have two vertical tabs, which were glued in using TAP Acrylic Cement (TAP Plastics, Mountain View, CA) and provide placement guides for proper rhizotron insertion. The bin tops had two posts in diagonally opposite corners, which were connected with sticky copper tape (McMaster-Carr #76555A641) placed on the underside of each bin top to create a conductive path that the robotic arm used to find each bin and calculate the location of each rhizotron. Every post was made from an M5 stainless steel button head screw (McMaster-Carr #92095A484),

35 mm long – 13 mm OD 18-8 Stainless Steel Unthreaded Spacer for M5 screws (McMaster-Carr, #92871A276), 25 mm long – 10 mm OD 18-8 Stainless Steel Unthreaded Spacer for M5 screws (McMaster-Carr, #92871A093), and an M5 stainless steel lock nut (McMaster-Carr #93625A200). The screw went through the larger spacer, through the bin top, through the smaller spacer, and everything was held together by the nut. Each post extended down and into a 10 mm hole precisely positioned at the edge of each black box.

All build files for the rhizotrons and holders can be found at https://doi.org/10.5281/zenodo.6694558.

## Biological components

### Six accessions

Bay, Col-0, Est, Sha, Tottarp, and Tsu-0 were transformed using the floral dip method (*Zhang et al., 2006*) with the *ProUBQ10::LUC2o* reporter, which consists of the UBQ10 promoter region and first intron amplified from Col-0 genomic DNA, the plant-codon-optimized LUC2 coding sequence, and a plasma membrane-localized mCherry coding sequence driven by the 35S promoter (additional details in Rellán-Álvarez et al., 2015). Positive transformants were selected under a fluorescent dissecting scope (Leica M165 FC) using the mCherry marker visible in the mature seed. Bay, Col-0, Est, Sha, and Tsu-0 seeds were carried through to the T3 generation. Tottarp was used in the T2 generation.

### Diversity panel

*A. thaliana* accessions were ordered from the Arabidopsis Biological Resource Center (ABRC: CS78885; *Supplementary file 2*). For vernalization, plants were grown in pots filled with Pro-Mix PGX soil (Premier Tech, Canada) and placed in a growth cabinet with long day conditions (16 hr light, 8 hr dark) at 10°C and 50% humidity using fluorescent bulbs with a light intensity of about 200 µE m$^{-1}$ s$^{-1}$. Upon flowering, plants were transformed as described above and then moved to a 22°C greenhouse with long day conditions and 150–250 µE m$^{-1}$ s$^{-1}$ light intensity. Transformation of the 192 initial accessions yielded 187 positive T1 lines, 171 of which could be confirmed in the T2 generation. Ten positive transformants from each of the T2 lines were screened on plates for strong luminescent signal once primary root length reached approximately 1.5 cm, usually around 2 weeks, but up to 26 days. Seedlings with strong root signal were given a line identification letter, then transferred to pots, and grown in the 10°C growth cabinet until inflorescences emerged (27–109 days after transfer to soil depending on the accession), at which point all plants were photographed and moved to a 22°C greenhouse until seeds could be harvested. Ultimately, 148 lines yielded at least three independent lines with strong root luminescence. Before starting an experiment, seeds were again screened for positive transformants using the mCherry seed coat marker.

## Growth method

### Rhizotron preparation

Rhizotron preparation was done as described in Rellán-Álvarez et al., 2015, with minor modifications: a polycarbonate plastic sheet with 8 mm holes and a sheet with the black rhizotron top and metal hooks were each laid on the table. Since the acrylic rhizotron top extends from the top in every direction, those sheets were laid down on the building surface with the acrylic piece and hooks lying just-off the edge of the surface and the screws pointing up, which ensured that the sheet lay flat. To prevent excess buildup of soil, the spacers were inserted into the sheets with 8 mm holes, and 'flags' (spacers modified with small pieces of material at the top to cover the bottom part of the hook and screw, Flags-final.eps) were inserted into the sheets with the attached tops. Using an electric paint sprayer (Wagner Spraytech Control Spray Double Duty HVLP Sprayer Model #0518050), a mist of water was applied to the sheets. Peat based Pro-Mix PGX soil (Premier Tech, Canada) was gently sifted over the sheets using a 2 mm sieve (US Standard Sieve Series N˚10) and excess soil was gently shaken off. The mist, sift, shake procedure was done for a second time, which creates a two-layer thick surface of soil thin enough to see a small amount of light through when held up to a lightbulb or window. To prevent soil from falling out, a folded and trimmed piece of paper towel was moistened then placed at the bottom of each rhizotron. The spacers and flags were removed and the flags were replaced with clean spacers. The two sheets are carefully placed together and rubber U-channels were slipped on to each side. A small handful of sifted soil was placed into the groove in the rhizotron top and gently pushed

down. Completed rhizotrons were hung in the boxes and, due to the increased size of the boxes, 3 L of water were added to the bottom of each box.

## Sowing and growth

Two transfer pipettes (each ~2 mL) of Peters Professional 20-20-20 General Purpose fertilizer were added to each rhizotron immediately after building. Fertilized rhizotrons in filled boxes were covered and left to soak overnight. Seeds were stratified for 2 days at 4°C in distilled water. Three seeds were sown in the center of each rhizotron using a p200 pipette (Eppendorf) and a unique barcode was placed on the rubber U-channel. Rhizotrons were sprayed down with water and a clear acrylic sheet was placed on top of the box then sealed with packing tape to create a humid environment. Three days after sowing, clear sheets were unsealed and rhizotrons were watered with two transfer pipettes of water. The following day, the clear sheets were removed and rhizotrons were watered again. Rhizotrons were watered with two pipettes of water until 9 days after sowing. Plants were thinned to one central plant 5 days after sowing. Plants were grown under long day conditions (16 hr light, 8 hr dark) at 22/18°C (day/night) using LED lights (Valoya, C-series N12 spectrum) with a light intensity of about 130 µE m$^{-1}$ s$^{-1}$. For the six accessions, plants were grown until 9 or 21 DAS and then were imaged every day until 31 DAS. Five replicates of each accession were grown for each treatment and plants were watered with luciferin every 6 days. For the diversity panel, the population was grown up, as a whole, six times, thus having temporal population replicates rather than internal replicates in order to reduce variation effects of growth condition. Since we imaged the set of 93 accessions in two parts (parts 1 and 2), we mostly split the accessions alphabetically and added two Col-0 controls per imaging session. The rhizotrons of each imaging session were then swapped once between boxes to avoid rhizotron building effects. For the second imaging session (part 2), we made sure to include a Col-0 control to the bin that grew in the shelf outside of GLO-Bot (part 2, Bin D) to examine position effects. After the first randomization, accessions were always planted in the same locations relative to each other (*Figure 4—figure supplement 2*) so the population was treated identically in each replicate. Plants were grown on shelves in the growth chamber next to the imager until 14 DAS. Subsequently, they were transferred into the imager and imaged every other day from 14 to 28 DAS with luciferin additions every 6 days.

## Plant imaging

On each imaging day (timing depending on experiment), GLO-Bot loaded each rhizotron into the imager then closed the door and triggered µManager to capture 5 min exposures on each side of the rhizotron. Inside the imaging system, there are two cameras on top of one another, with one taking an image of the top part of the rhizotron and the other, the bottom part of the rhizotron. After one side was imaged, the rhizotrons were rotated using a Lambda 10-3 Optical Filter Changer (Sutter Instrument, Novato, CA) and the other side of the rhizotron was imaged using the top and bottom camera, yielding four images per rhizotron. If it was the first imaging day or a designated luciferin day (every 6 days), GLO-Bot added 50 mL of 300 µM D-luciferin (Biosynth International Inc, Itasca, IL) to the top of each rhizotron immediately before loading the rhizotron into the imager. Before every replicate, a blank image was taken to account for background noise within the imager.

## Image analysis

Initial root image preparation to combine and align the four raw images was done as described in Rellán-Álvarez et al., 2015. Initial background noise removal was done using remove.background.ijm. By manually opening the blank image for the respective replicate and then running this macro, the open blank image will be subtracted from all of the files in the folder being processed by the macro. The ImageJ plugin Template Matching and Slice Alignment was run as instructed in the manual to register all root images. Images are then run through the macro denoise_overlay.ijm to remove additional noise and add images together, which maximizes signal intensity throughout the root by: (1) subtracting 1.5 from all pixels to remove pixels with low values; (2) multiplying all values by 3, which greatly amplifies pixels with high values; (3) subtracting 8 from all values, which again removes pixels with low values (i.e. those that did not get amplified); (4) running the ImageJ command Despeckle, which replaces every pixel with the median value in its 3×3 neighborhood (*Schindelin et al., 2012*; *Schneider et al., 2012*); (5) setting and using a threshold to generate a binary image; (6) running the

ImageJ commands Despeckle, Erode, and Dilate to further remove small pieces for noise; (7) using the ImageJ command Analyze Particles to get rid of small particles; and (8) merging the cleaned image with that of the previous day using the ImageJ command Add Create. This macro can be adjusted using the subtract-multiply-subtract sequence to change how signal intensity is amplified by manually testing the values on a handful of images.

Next, macros invert.ijm and tip_tracking.ijm are run to prepare images for GLO-RIAv2 (https://doi.org/10.5281/zenodo.5574925). This macro works by: (1) opening two consecutive images; (2) using the ImageJ command Dilate twice on the image from the earlier time point; (3) using the ImageJ command Subtract Create to subtract the earlier day from the later day. By dilating the earlier root system, we are able to ensure complete subtraction from the later time point and, therefore, only analyze the new root growth. Again, the thresholding in the new growth isolation can be refined if needed. Images were then run through GLO-RIAv2 and downstream analysis was performed in R (*R Development Core Team, 2019*) for additional cleanup and calculating traits. Initial formatting and combination of GLO-RIA output files is done using 1-format.Rmd. This file uses the raw output, the 'key', which contains information about the plants, and the experimental start date to calculate relative imaging times (very important for comparing multiple experiments), extract identifying information for each rhizotron, and, for the local files, calculate the root angles with respect to gravity using trigonometric calculations based on whether the raw output is greater than or less than 90°. Although we re-calculate the angles from 0 to 90°, having the 0 to 180° angles tells us which way the vector is pointing and which trigonometric function should be used to calculate the end point for the root segment. It should be emphasized that angles are with respect to gravity and not with respect to the parent root. Formatted files are run through 2-clean.R to remove stray particles. This process works by (1) computing the distances between all root x-y points in an image; (2) using hclust(method = "single") to cluster those distances; (3) split the cluster tree into two groups using cutree(); (4) calculating the distance between the groups and determining if it is greater or less than the predetermined proximity maximum; (5) if the distance is smaller than that distance the cluster is kept, if it is larger than the distance, cluster with the larger minimum distance to the center-x is kept and the other cluster is further examined; (6) in the cluster being examined, the cluster is again tested for whether the distance between the clusters is far too large (nearly twice that of the initial proximity maximum) or whether there are less than four points in the cluster, which would indicate the cluster is likely noise. If either of these are true, the cluster is discarded. If not, the cluster is further examined to test if the points are linear, which would indicate the cluster could be a root segment. The above sequence is done for 1 day, then the next day is added in and the process begins again, and this continues until only the 'true root' segments remain. Parameter adjustments in this file primarily depend on the strength of the root signal and can be optimized through trial and error. The 2-clean.R script outputs a text file describing which clusters were kept and removed, a pdf for each rhizotron showing all of the removal steps, two csv files, clean_removed_points.csv and clean_true_root.csv, which list all of the points that were either kept or removed, as well as two initial trait files: clean_ROIs.csv and clean_traits.csv. Having these 'true root' segments allows for calculation of traits at the new growth level as well as the whole root level. Some of these calculations are done during the root cleaning process, while others are re-calculated later in 3-summarize_traits.Rmd.

Images were manually curated to eliminate those with very high levels of noise or from occasional camera malfunction. Additionally, phosphorescent ('glow-in-the-dark') stars in the upper right corner of some images were manually cropped out. Initially, these stars were placed in the images to aid image alignment, but instead generated too much noise. For the GLO-Roots diversity panel, a total of 34 images were removed (*Supplementary file 6*). Those images were replaced by an image of the previous day to ensure the correct number of images for GLO-RIAv2 analysis.

## Trait processing

Raw trait measurements were converted into single values for each genotype in the software R (*R Development Core Team, 2019*) using the script mcmcglmm_best.R. This script is run for each trait, and the user inputs the raw phenotypic data for each trait. Several models were tested, including a second-order polynomial for time, different error correction factors, and models with and without kinship-informed random factor. Each model was examined for parameter convergences using the deviance information criteria (DIC), a Bayesian model version of Akaike information criterion (AIC).

The selected model was chosen based on speed, simplicity, and generalizability. It should be noted that some traits may benefit from a higher order function. The final model had the form:

$$y = \beta_0 + \left(\beta_1 + u_{genot}\right) t + u_{genoi} + u_{rep} + e$$

where the trait (y) and time (t) were fitted as fixed effects, genotype ($u_{genoi}$), replicate ($u_{rep}$), and the time and genotype interaction ($u_{genot}$) were treated as random factors. The effects capture the average deviation for each genotype from the mean of the population. The script runs the data through a null model, which is a model run with no parameters and provides the baseline comparison for future models using the same input data, and then runs the previously described mixed model to estimate fixed effects by running 1,000,000 iterations in a Monte Carlo Markov chain (MCMC) and 200,000 burn-in using the MCMCglmm R package (*Hadfield, 2010*). Since MCMCglmm utilizes an MCMC walk, the model has inherent stochasticity and can produce slightly different values each time it is run. For reproducibility, we set a 'seed number' for the script then ran the model three times for each trait then selected the model with the best DIC.

Once the model is fit, we extract the genotype-specific parameters which capture differential starting values of a trait and differential trajectories over time. With these intercept and slope parameters, we can calculate the value of the trait for each genotype at any desirable time point, to understand the genotype's trait behavior. We do this for the time points 0, 48, 96, 144, 192, 240, 288, and 336 hr to match our imaging frequency. These fitted genetic values are termed our 'fitted values': the coefficients (intercept and slope) as well as the expected value at each time point for each genotype. Broad-sense heritability was calculated using standard approaches in MCMCglmm (*Hadfield, 2010*), which essentially quantify variation on the trait trajectories associated to genotype relative to the total (*Hadfield, 2017*).

In addition, once we extracted genotype-specific values of each trait (either the intercept or slope coefficient or the predicted value at a time point), we conducted GWAs using GEMMA and the AraGWAS database SNP matrix (*Zhou and Stephens, 2012*; *Togninalli et al., 2018*). GEMMA was run as a linear mixed model (lmm) and with a kinship matrix to correct for accession relatedness. Only SNPs with a minimum allele frequency greater than 0.05 were examined. Bonferroni threshold, which was calculated using the number of linkage blocks within our population as computed by PLINK, was used as the significant threshold (0.05/113831 LD blocks). Manhattan plots were generated in R.

## Acknowledgements

Work in the JRD lab was funded by the US Department of Energy's Office of Biological and Environmental Research (DE-SC0008769 and DE-SC0018277) and the Carnegie Institution for Science Endowment. TSL was funded by the National Science Foundation Graduate Research Fellowship and the National Institutes of Health Predoctoral Training Grant (T32GM007276). HL was funded by a German Research Foundation (DFG) research fellowship (LI 2776/1-1). MEA is funded by the Carnegie Institution for Science, an NIH Early Investigator Award (1DP5OD029506-01), and a DOE BER grant (DE-SC0021286). GL is funded by the German Research Foundation under Germany's Excellence Strategy, EXC-2070–390732324 (PhenoRob). Part of the computational analyses were done at the Carnegie High-Performance Computing clusters *Memex* and *Calc*.

We thank Peter Sand and Zoltan DeWitt from Modular Science for the design and maintenance of GLO-Bot. We thank Wolfgang Busch for advice on the initial selection of *Arabidopsis* accessions. We thank Chris Ballard from Sutter Instrument for maintenance of GLO1. We thank Theo van de Sande and Ismael Villa for maintenance of the growth room. We thank Michael Raissig and Josep Vilarrasa-Blasi for constructive suggestions on the manuscript.

# Additional information

## Funding

| Funder | Grant reference number | Author |
|---|---|---|
| U.S. Department of Energy | DE-SC0008769 | José R Dinneny |
| U.S. Department of Energy | DE-SC0018277 | José R Dinneny |
| National Institutes of Health | T32GM007276 | Therese LaRue |
| Deutsche Forschungsgemeinschaft | LI 2776/1-1 | Heike Lindner |
| National Institutes of Health | 1DP5OD029506-01 | Moises Exposito-Alonso |
| U.S. Department of Energy | DE-SC0021286 | Moises Exposito-Alonso |
| Deutsche Forschungsgemeinschaft | EXC-2070 - 390732324 | Guillaume Lobet |

The funders had no role in study design, data collection and interpretation, or the decision to submit the work for publication.

## Author contributions

Therese LaRue, Conceptualization, Data curation, Software, Formal analysis, Supervision, Funding acquisition, Validation, Investigation, Visualization, Methodology, Writing – original draft, Project administration, Writing – review and editing; Heike Lindner, Conceptualization, Data curation, Formal analysis, Supervision, Funding acquisition, Validation, Investigation, Visualization, Methodology, Writing – original draft, Project administration, Writing – review and editing; Ankit Srinivas, Investigation; Moises Exposito-Alonso, Software, Formal analysis, Supervision, Funding acquisition, Investigation, Methodology, Writing – review and editing; Guillaume Lobet, Resources, Software, Funding acquisition, Methodology, Writing – review and editing; José R Dinneny, Conceptualization, Resources, Data curation, Supervision, Funding acquisition, Investigation, Methodology, Writing – original draft, Project administration, Writing – review and editing

## Author ORCIDs

Therese LaRue  http://orcid.org/0000-0002-6805-5118
Heike Lindner  http://orcid.org/0000-0002-5913-9803
Moises Exposito-Alonso  http://orcid.org/0000-0001-5711-0700
José R Dinneny  http://orcid.org/0000-0002-3998-724X

## Decision letter and Author response

Decision letter https://doi.org/10.7554/eLife.76968.sa1
Author response https://doi.org/10.7554/eLife.76968.sa2

# Additional files

## Supplementary files

• Supplementary file 1. Analyzed root traits.

• Supplementary file 2. Accessions used for genome-wide association studies (GWAS).

• Supplementary file 3. Replicates with anomalies.

• Supplementary file 4. SNP list of 29 significant SNPs.

• Supplementary file 5. Broad-sense heritability of traits.

• Supplementary file 6. Removed images of the Growth and Luminescence Observatory for Roots (GLO-Roots) diversity panel.

• Transparent reporting form

## Data availability

GLORIAv2 is available through Zenodo, DOI: https://doi.org/10.5281/zenodo.5709009 https://doi.org/10.5281/zenodo.5574925. Image analysis pipelines and scripts are available through Zenodo, DOI: https://doi.org/10.5281/zenodo.5708430. RShiny App for exploring root system architecture of the six accessions is available through Zenodo, DOI: https://doi.org/10.5281/zenodo.6757675. RShiny App for exploring root system architecture of the diversity panel is available through Zenodo, DOI: https://doi.org/10.5281/zenodo.5708422. Imaging data and images are available through Zenodo, DOI: https://doi.org/10.5281/zenodo.5709009. General code for software operating robotics available: GitHub: https://github.com/rhizolab/rhizo-server. Rhizotron laser cutting files are available through Zenodo, DOI: https://doi.org/10.5281/zenodo.6694558. Previously published datasets used: WORLCLIM2: Fick SE, Hijmans RJ, 2017, https://worldclim.org/, https://doi.org/10.1002/joc.5086.

The following dataset was generated:

| Author(s) | Year | Dataset title | Dataset URL | Database and Identifier |
|---|---|---|---|---|
| LaRue T, Lindner H, Srinivas A, Exposito-Alonso M, Lobet G, Dinneny J | 2021 | GLO-Bot Data | https://doi.org/10.5281/zenodo.6757675 | Zenodo, 10.5281/zenodo.6757675 |

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
