## [Editor Report]

The authors present an automated system for phenotyping root system architecture based on bioluminescent roots resulting from a constitutively expressed luciferase transgene (GLO-Roots). They have developed a robotics-assisted phenotyping platform and an automated image analysis pipeline for high throughput analysis. An impressive array of 93 luciferase expressing *Arabidopsis thaliana* accessions provides a major resource for understanding the genetic basis for root system architecture variation under physiologically relevant conditions. The work will be of great interest to plant biologists and all those studying genetic variation in plants.

---

## [Decision Letter]

**Decision letter after peer review:**

Thank you for submitting your article "Uncovering natural variation in root system architecture and growth dynamics using a robotics-assisted phenomics platform" for consideration by *eLife*. Your article has been reviewed by 3 peer reviewers, and the evaluation has been overseen by a Reviewing Editor and Jürgen Kleine-Vehn as the Senior Editor. The following individuals involved in the review of your submission have agreed to reveal their identity: Niklas Schandry (Reviewer #1); Larry York (Reviewer #3).

As you see from the reviewers' comments they (and myself) are excited about your manuscript, although it requires some revision and clarifications to become acceptable for *eLife*. Please find the reviewer comments below. We have discussed the experiment with 6 accessions (mentioned by reviewer #3) and we agree that it should stay in the manuscript but that it also causes some distraction from the main point. We suggest to more strongly focus the text on these results to better integrate them with the next chapter describing results from the full panel. Otherwise, we consider all reviewer comments to be important to improve the manuscript.

*Reviewer #1 (Recommendations for the authors):*

Comments on GWAS:

– I noticed that sometimes, the data cleaning procedure appears to retain pixels which are not part of the root system (e.g., Ting-1 in GWA-5; Tomegap2 in GWA-1 and GWA-5 or Dra2-1 in GWA3).

– The authors use “depth” to refer to the vertical distance covered by the overall root system, and “length” to refer to the sum of the length of all vectors that represent the root system. In the extreme case of a single main root growing straight down depth would be equal to cumulative length and depth should never be larger than cumulative length, yet there are cases where depth is larger than the total length of the root (Kz9, GWA3 and Kor3 GWA2). The length measurement in several accessions suffers from unconnected vectors, while the overall shape of the root system is probably correct (depth, width, hull). See for example accession Ge-0 GWA3, GWA5 or GWA6, where the root appears to be made up of many very short vectors with gaps. This raises the question if the length variable is suitable for quantitative genetic analysis. I also wonder how this issue affects the accuracy of the vector angles with respect to the true root system.

– Breeding values were computed for the quantitative genetic analysis. It should be clarified whether “breeding value” refers to the estimated model coefficients (p. 8), or to the fitted values (p. 54). All traits were overall using the same formula. It is not really clear why the authors opted for this approach instead of using different models for the different traits. Some traits, such as width (see Figure S5) do not display a linear relationship with time.

Average angle per day appears to extremely dynamic and I am not sure if the breeding values for this trait are meaningful with regards to the original data, raising the question if the fitted values for this trait are a good input for GWAS. Is it possible that the overall observation that p-values of some SNPs decrease over time simply reflects the linear relationship with time specified in the model, as fitted values increase over time?

What is the advantage of using fitted values at different timepoints over using the model coefficients for each accession as the input?

Since the model assumes a linear relationship, which appears to not capture the dynamics of all traits, would it not be reasonable to simply compute the trait mean per day per accession and use these values as input for GWAS, instead of using fitted values?

*Reviewer #2 (Recommendations for the authors):*

1. While the GLO-Bot system is using potting mix, it will still be very different from soil grown roots. With GLO-roots, the roots are grown in a thin sheet of soil (essentially restricted to 2 dimensions) and the roots will become constrained by the width of the system as well. This will most likely reduce differences between shallow rooting accessions and intermediate accessions. It will be important to discuss the constraints of the system and discuss whether any time point will be closer to a 3D soil growth environment. One way how this might be elucidated would be to look at broad sense heritability at different time points. Such an approach might also indicate an optimal timepoint for the GWAS and the correlations with environmental variables.

2. Page 8: "For each of these traits, we used a generalized linear mixed model in the R package MCMCglmm to account for replicate and block effects noises and estimate the "breeding value" of each accession for a given trait (Wilson et al. 2010; Mrode 2014); (Hadfield 2010) (Supplemental Figure 5). The denoised trait values for each genotype at 0, 48, 96, 144, 192, 240, 288, and 336 hours after the start of imaging (equivalent to 14, 16, 18, 20, 22, 24, and 28 DAS) were used for further analyses."

It remains a bit unclear what the process was (how were the block effects defined, etc.) and what the rationale was to use "breeding value". Also, most readers will not know what a "breeding value" is. Are the "denoised" trait values breeding values? Why do the authors think that the raw trait values are not helpful for further analysis (after all, many of the BSH seems reasonable)?

3. Page 9ff: The candidate gene follow up is very incomplete. For instance, the authors show in their Manhattan plots that some associations are linked to other SNPs in that region. It therefore would be good to also include genes in proximity of the linked SNPs in the table. Also, the SNPs identified with a GWAS are not necessarily the causal SNPs (in particular, this is an issue if not the full genome sequence was used for GWAS). The causal SNPs could be linked or partially linked to the GWAS SNPs (e.g. due to allelic heterogeneity). It would therefore be helpful if for the genes discussed in the text, the genomic region could be shown with haplotypes or just using the 1001 genome project SNPs in accessions with alternative GWAS SNPs.

4. Page 5: Root Angle output is unclear (also in the Suppl. Table). This is important as it is a major trait in the manuscript. Is this the average of all roots? Do all roots need to be detected for that? What are the two points that are used to measure the angles? What is a root segment? How many are considered?

5. Page 6: How many root systems were compared with SmartRoot? Is each point in Figure 2B an entire root system? Is the comparison similarly accurate for very different RSAs? What is the advantage of GloRoot vs Smartroot (i.e. how much time does it take to trace a GLO-bot RSA manually)?

6. The trait data is not deposited with the manuscript or on a public repository. This would be very important to allow the community to build on the work of the authors.

*Reviewer #3 (Recommendations for the authors):*

The comments below may be helpful to the authors.

Since there weren’t line numbers, I give page numbers below and tried to ‘quote’ key words that should allow the authors to find what my comments pertain to.

1: ‘above ground’ is not hyphenated here but is elsewhere.

1: 'climate variables of the accessions respective origins' or something as to not be confused with growth conditions, etc.

2: 'between' should be among.

2: "Most RSA studies…" – can the most here really be substantiated? If not, "many" may suffice. I am not sure that most are in gel or in Arabidopsis.

3: rhizotrons are often called rhizoboxes. If rhizotron is preferred, perhaps "(or rhizobox)" can be added at first mention to orient some readers.

5: 'detection and translation of subsequent images' – I believe this describes what is commonly called registration so the word may be useful to add here for clarity.

5: 'invert the images' may be expressed as 'invert the colors of the images' to be precise.

5: 'root segment' needs defined.

6: 'series of vectors' – seems like RSML and some would ask if RSML is supported.

6: 'half of which' could be confused for accession or replicate (I know it's replicate) but consider saying "where five replicates" instead.

6: for the 6 accessions, you can add a bit more quantification by using a calculation of heritability or repeatability. You could have also used the additional replication for power analysis to determine the need for replications numbers, for example boot strapping from 2 – 10 reps to see if replicates improve measures or heritability.

7: in the last paragraph, what type of randomization used should be specified (maybe in methods, I don't like this format where some methods are in results but not all, but I know it's because results are first). On page 8 it is mentioned that an R package is used for block. I think this is leading me to conclude the results simply have to methods mixed in.

8: So is it right that a subset of days are used? Why?

8: PCA over time implies time dependent traits are just sized related.

9: ‘fitted values’ could be clarified.

10: ‘published bioclimatic variables’ could include for clarity ‘…from the origin sites of the respective accessions.’

11: ‘progress in machine-learning’ – I remember I think one of the original rationales for this system was to image on both sides for a more complete root system? Was that in the original paper? Is that done here? With regards to bioluminescence – is it still clear that this method is advantageous to RGB imaging or other forms of imaging? For example, the Smith et al. 2020 paper shows many examples of identifying roots from complex backgrounds. The RootPainter software has been used for a rhizobox study:

https://www.biorxiv.org/content/10.1101/2021.08.24.457510v1

Can the possible advantages be discussed, especially if there are specific examples from images that show color imaging alone is not sufficient?

11: sample sizes could be sample numbers I think.

Figure 1 – scale needed in all panels.

48: Not clear why Materials and methods are in supplemental?

51: "Accessions were always planted in the same locations relative to each other so the population was treated identically in each replicate." Strictly speaking – this is a bad choice, somehow you decided to intentionally NOT randomize which goes against statistical training and a strict reviewer may say it compromises the experimental design where all the measurements could be artifacts of position. Is there a good reason to not randomize? It seems odd in what otherwise seems like such careful design.

51: 'four raw images' – root and shoot from each side is four? Maybe make clear.

51: what is the 'correct blank images' – is this process entirely automated? Sounds like a user walks the processing through several fairly automated steps but not quite automated from image to data it seems.

---

## [Author Response]

As you see from the reviewers' comments they (and myself) are excited about your manuscript, although it requires some revision and clarifications to become acceptable for eLife. Please find the reviewer comments below. We have discussed the experiment with 6 accessions (mentioned by reviewer #3) and we agree that it should stay in the manuscript but that it also causes some distraction from the main point. We suggest to more strongly focus the text on these results to better integrate them with the next chapter describing results from the full panel. Otherwise, we consider all reviewer comments to be important to improve the manuscript.Reviewer #1 (Recommendations for the authors):Comments on GWAS:– I noticed that sometimes, the data cleaning procedure appears to retain pixels which are not part of the root system (e.g., Ting-1 in GWA-5; Tomegap2 in GWA-1 and GWA-5 or Dra2-1 in GWA3).

We thank reviewer #1 for this comment. It indeed happens that the data cleaning procedure does not remove all background pixels. The algorithm cannot distinguish background pixels from root pixels and it needed the mentioned adjustments to retain as much root signal as possible and remove as much background noise as possible. Since we wanted all images to be denoised in the same way, we had to compromise with some images showing remaining background noise. A list of images with these problems has been curated and is included in Supplementary File 3. Certain traits, such as average angle, have been weighted by vector length so tiny measurements from the noise do not have a strong influence on the trait value. Further, we expect noise to not be systematically associated with certain genotypes, rather it will be randomly scattered across replicates within a genotype and across all the genotypes. This measurement error would therefore result in lower heritability estimates, making the conclusions we draw naturally more conservative.

– The authors use “depth” to refer to the vertical distance covered by the overall root system, and “length” to refer to the sum of the length of all vectors that represent the root system. In the extreme case of a single main root growing straight down depth would be equal to cumulative length and depth should never be larger than cumulative length, yet there are cases where depth is larger than the total length of the root (Kz9, GWA3 and Kor3 GWA2). The length measurement in several accessions suffers from unconnected vectors, while the overall shape of the root system is probably correct (depth, width, hull). See for example accession Ge-0 GWA3, GWA5 or GWA6, where the root appears to be made up of many very short vectors with gaps. This raises the question if the length variable is suitable for quantitative genetic analysis. I also wonder how this issue affects the accuracy of the vector angles with respect to the true root system.

We thank reviewer #1 for addressing this point. They are correct, the total length measurements come with caveats due to the nature of the unconnected vectors and therefore the unaccounted length between detected root segments. This caveat about the length measurement and other potential caveats have been included in the text on page 6/7. For the vector angles, each vector angle is weighted by the length of the segment, so angles computed from very small short segments are weighted differently than longer connected segments. While it is true that the angles for fully connected roots could differ, this is the most objective and robust approach we could devise and, given the high heritability of the trait, we believe the signal is well captured.

– Breeding values were computed for the quantitative genetic analysis. It should be clarified whether “breeding value” refers to the estimated model coefficients (p. 8), or to the fitted values (p. 54). All traits were overall using the same formula. It is not really clear why the authors opted for this approach instead of using different models for the different traits. Some traits, such as width (see Figure S5) do not display a linear relationship with time.Average angle per day appears to extremely dynamic and I am not sure if the breeding values for this trait are meaningful with regards to the original data, raising the question if the fitted values for this trait are a good input for GWAS. Is it possible that the overall observation that p-values of some SNPs decrease over time simply reflects the linear relationship with time specified in the model, as fitted values increase over time?What is the advantage of using fitted values at different timepoints over using the model coefficients for each accession as the input?Since the model assumes a linear relationship, which appears to not capture the dynamics of all traits, would it not be reasonable to simply compute the trait mean per day per accession and use these values as input for GWAS, instead of using fitted values?

We thank the reviewer for the point regarding breeding values. We realize this was not explained in enough detail. The reviewer is correct about the estimated per-genotype trait deviations and the terminology in the text was unclear. For the sake of clarity, the terminology has been changed to only use “fitted values''. Using our temporal model, the slope and intercept coefficients were used to calculate the expected genotype value at any timepoint. The text on page 8 (now page 9) was clarified.

Both using the accessions’ coefficients as well as the predicted value at a given timepoint are fair and interesting approaches to study the differences in root traits among ecotypes, and each provides different conclusions. Simply using the coefficients for GWA will tell us about the genes responsible for the main trends in the data, but these are less intuitive. Conducting GWA on the predicted values at each timepoint gives us an understanding about when a trait is most relevant, which is an important conclusion of our efforts to create a longer time series of trait data than previously available.

Another question from the reviewer regards the different possible functions used to model traits; this is a matter of statistical tractability and trade-offs between different models. While higher order polynomials or sigmoid curves may fit well or even better than a linear relationship of the trends, these models were not tractable in our experimental and analysis design. Let us give some background: To make the most of the data, the approach we took is that of time series analyses of Bayesian generalized linear mixed models where we can include experimental design factors to control for batch effects and at the same time fit genotype factors to estimate genotype-specific trends (breeding values of the coefficients). This is advantageous because we can control for noise while using all the replicates and all the time points to infer genotypic effects (this is a common approach in complex breeding programmes and designs with temporal datasets to extract breeding values, see: http://cran.nexr.com/web/packages/MCMCglmm/vignettes/CourseNotes.pdf). The limitation is that when temporal functions (polynomial, etc.) become more complex, the model requires fitting extra parameters (number of genotypes * new function parameter). This makes it infeasible to fit very complex models, and indeed when we tried using MCMCglmm the models often did not converge to a solution. One may be tempted to fit a polynomial model for all the data points for each genotype separately, which would be feasible. However, this approach would not enable calculating breeding values for accessions, and would lack control for batch effects. Given these trade-offs of model selection, we decided to sacrifice complexity of the time series function in favor of better control for experimental design.

Reviewer #2 (Recommendations for the authors):1. While the GLO-Bot system is using potting mix, it will still be very different from soil grown roots. With GLO-roots, the roots are grown in a thin sheet of soil (essentially restricted to 2 dimensions) and the roots will become constrained by the width of the system as well. This will most likely reduce differences between shallow rooting accessions and intermediate accessions. It will be important to discuss the constraints of the system and discuss whether any time point will be closer to a 3D soil growth environment. One way how this might be elucidated would be to look at broad sense heritability at different time points. Such an approach might also indicate an optimal timepoint for the GWAS and the correlations with environmental variables.

We thank reviewer #2 for this very important point. We agree with the reviewer that the dimensions of the growth vessel will impact the architecture of the root system, as will soil type. While measurements of root growth in large volumes of natural soil will improve the ecological significance of the trait measurements, we are not aware of data supporting the view that this will necessarily affect the heritability of these traits. Likely, heritability will be affected by the complexity of the environment that roots grow; conditions such as agar gel will likely exhibit tighter relationships between genotype and phenotype because there is less heterogeneity in the environment. This fundamentally differs from more natural soil conditions where heterogeneity in the environment is intrinsic to the growth condition. We believe that the relationship between genotype and phenotype has greater ecological significance when measured using the GLO-Roots system due to the more physiologically relevant ways in which light, water and nutrient availability interact with the root system and processes such as light, transpiration and gas exchange impact the plant as a whole. Certainly, as soon as the roots touch the edges of the rhizotron, traits like root depth and root width will lose significance, however traits like average angle or length per day will still show differences until the rhizotron is filled with roots.

We conducted an analysis of heritability over time for our traits, although we had not included it in the text. One could conduct MCMCglmm for each timepoint. Another alternative is to study the proportion of variance explained (PVE) of each trait by timepoint by the genome in the GWA analyses. GEMMA provides such PVE, a parameter that is also often called Chip heritability or narrow sense heritability, as the kinship matrix, built to control for genotype relatedness, captures additive differences. Similarly as Figure 6A, we can see how the PVE changes over time, see Author response image 1. It is interesting to note the relatively high amount of variance explained for traits such as weighted average angle and weighted average angle per day. These data are interesting leads to continue working on in the future.

**Author response image 1. sa2fig1:** 

2. Page 8: "For each of these traits, we used a generalized linear mixed model in the R package MCMCglmm to account for replicate and block effects noises and estimate the "breeding value" of each accession for a given trait (Wilson et al. 2010; Mrode 2014); (Hadfield 2010) (Supplemental Figure 5). The denoised trait values for each genotype at 0, 48, 96, 144, 192, 240, 288, and 336 hours after the start of imaging (equivalent to 14, 16, 18, 20, 22, 24, and 28 DAS) were used for further analyses."It remains a bit unclear what the process was (how were the block effects defined, etc.) and what the rationale was to use "breeding value". Also, most readers will not know what a "breeding value" is. Are the "denoised" trait values breeding values? Why do the authors think that the raw trait values are not helpful for further analysis (after all, many of the BSH seems reasonable)?

We realize that the rationale of the MCMCglmm approach was not described in enough detail. We have now included a much more extended version in the trait processing section of the methods. Further, we clarified the description highlighted in this comment on page 9 to be more consistent with the rest of the text. We have removed the term “breeding value” and instead use the general term “fitted values” to represent all of the fitted genetic values for each genotype: the slope, intercept, and the expected values at each time point for each genotype.

3. Page 9ff: The candidate gene follow up is very incomplete. For instance, the authors show in their Manhattan plots that some associations are linked to other SNPs in that region. It therefore would be good to also include genes in proximity of the linked SNPs in the table. Also, the SNPs identified with a GWAS are not necessarily the causal SNPs (in particular, this is an issue if not the full genome sequence was used for GWAS). The causal SNPs could be linked or partially linked to the GWAS SNPs (e.g. due to allelic heterogeneity). It would therefore be helpful if for the genes discussed in the text, the genomic region could be shown with haplotypes or just using the 1001 genome project SNPs in accessions with alternative GWAS SNPs.

We thank the reviewer for this point. We know the SNPs associated with the trait are not necessarily the causal ones. We have included the list of all genes within a 20 Kb region of the significant SNPs (10 kb upstream and 10kb downstream from the SNP) in Supplementary File 4 also listing the reference and alternative alleles. This has been clarified in the text, as well.

4. Page 5: Root Angle output is unclear (also in the Suppl. Table). This is important as it is a major trait in the manuscript. Is this the average of all roots? Do all roots need to be detected for that? What are the two points that are used to measure the angles? What is a root segment? How many are considered?

Root angles are computed on sections of roots visible in the image (aka “segments”). These segments do not always correspond to a biological structure, but rather the visible readout of the root system structure within the GLO-Bot platform. These root system structures visible on the GLO-Bot images are indeed disconnected in many locations, due to variability in the reporter’s intensity and the presence of physical obstructions such as soil particles. We actually used this disconnection to our advantage to compute more faithful root angle measurements within the image, as we use the “segments” as individual elements (see Author response image 2). We have added a detailed clarification to the text on page 6.

5. Page 6: How many root systems were compared with SmartRoot? Is each point in Figure 2B an entire root system? Is the comparison similarly accurate for very different RSAs? What is the advantage of GloRoot vs Smartroot (i.e. how much time does it take to trace a GLO-bot RSA manually)?

The growing root systems of 5 plants were traced with SmartRoot. Since each root was imaged over a series of days and occasionally imaged multiple times within one day each plant has between 15 and 22 root images. In total, 95 root images were traced. Each point in the plot is a time point. Manually tracing a single GLO-bot RSA image with SmartRoot can take days, depending on the size of the root system. SmartRoot relies on continuous structures to follow the root. Due to the nature of the GLO-bot RSA images, this becomes a very tedious process of tracing each root system. We added more details to the main text and the figure legend of Figure 2B, stating: “Manual SmartRoot tracing of a single GLO-Roots image can take multiple days, thus making analysis of large-scale time course experiments unfeasible.”

6. The trait data is not deposited with the manuscript or on a public repository. This would be very important to allow the community to build on the work of the authors.

The trait data is available on Zenodo (https://doi.org/10.5281/zenodo.5709009) as stated at the end of the manuscript. A link has also been inserted into the text for clarity.

Reviewer #3 (Recommendations for the authors):The comments below may be helpful to the authors.Since there weren’t line numbers, I give page numbers below and tried to ‘quote’ key words that should allow the authors to find what my comments pertain to.1: ‘above ground’ is not hyphenated here but is elsewhere.

We thank reviewer #3 for this comment and corrected this mistake. It now reads; ”The plant kingdom contains a stunning array of complex morphologies easily observed above-ground, but more challenging to visualize below-ground.”

1: 'climate variables of the accessions respective origins' or something as to not be confused with growth conditions, etc.

We have added in ‘of the accessions’ respective origins’ for clarity.

2: 'between' should be among.

Thank you, we changed “between” to “among”.

2: "Most RSA studies…" – can the most here really be substantiated? If not, "many" may suffice. I am not sure that most are in gel or in Arabidopsis.

We thank the reviewer for this valuable input and agree that “most” should be replaced by “many”.

3: rhizotrons are often called rhizoboxes. If rhizotron is preferred, perhaps "(or rhizobox)" can be added at first mention to orient some readers.

Thank you for this suggestion. We prefer rhizotrons to avoid confusion with the growth boxes that hold the rhizotrons in our setup. We added “rhizoboxes” to the text. It now reads; “In addition to GLO1, the frame contains a growth area for seven black polyethylene growth boxes (Figure 1 B), each of which holds 12 rhizotrons (also known as rhizoboxes) arranged in a two by six grid (Figure 1 C, Figure 1—figure supplement 1B).

5: 'detection and translation of subsequent images' – I believe this describes what is commonly called registration so the word may be useful to add here for clarity.

We agree with the reviewer and adapted the text. It now reads: “This plugin works by user selection of a region present in all images followed by automated detection and translation of subsequent images, a method that is also known as image registration.”

5: 'invert the images' may be expressed as 'invert the colors of the images' to be precise.

For clarity, this has been revised to: “…invert the gray-scale values of the images, thus making the root system itself black and the background white for downstream whole root system analyses.”

5: 'root segment' needs defined.

Root angles are computed on sections of roots visible in the image (aka “segments”). These segments do not always correspond to a biological structure, but rather the visible readout of the root system structure within the GLO-Bot platform. These root system structures visible on the GLO-Bot images are indeed disconnected in many locations, due to variability in the reporter’s intensity. We actually used this disconnection to our advantage to compute the root angle distribution within the image, as we use the “segments” as individual elements (see figure in comment 2.4). This also has been clarified in the text on page 6.

6: 'series of vectors' – seems like RSML and some would ask if RSML is supported.

We choose not to, since in the analysis we lose most of the topological information that is central in the RSML structure.

6: 'half of which' could be confused for accession or replicate (I know it's replicate) but consider saying "where five replicates" instead.

Thank you for this valid concern, we changed the text accordingly. It now reads; ”Each accession was grown in ten replicates, where five replicates were imaged daily from 9 days after sowing (DAS) until 31 DAS and the other five replicates from 21 – 31 DAS.”

6: for the 6 accessions, you can add a bit more quantification by using a calculation of heritability or repeatability. You could have also used the additional replication for power analysis to determine the need for replications numbers, for example boot strapping from 2 – 10 reps to see if replicates improve measures or heritability.

We thank reviewer #3 for these comments. We calculated heritability for the 6 accessions but chose not to include it since the small number of genotypes precludes confident heritability determination. We recognize the point about determining replicate numbers, however this was not something we did at the time. While the 6 accessions provided insight into some of the methods (e.g. luciferin watering speed and frequency, optimal imaging time window, and ground truthing image analysis methods), they did not inform factors such as replicate number in the experiment. Based on reviewer comments, we are choosing to not only emphasize the 6 accessions as a foundation for the diversity panel, but also highlighting them as an alternative example of how the GLO-bot system can be used (i.e. few accessions vs many accessions, high imaging frequency vs every other day imaging, raw data vs. fitted data, etc.).

7: in the last paragraph, what type of randomization used should be specified (maybe in methods, I don't like this format where some methods are in results but not all, but I know it's because results are first). On page 8 it is mentioned that an R package is used for block. I think this is leading me to conclude the results simply have to methods mixed in.

The 93 accessions of the GLO-Roots diversity panel were selected balancing genetic similarity and environmental variation along with reporter intensity, and the number of independent transformation events recovered. Since we imaged the set of 93 accessions in two parts (part 1 and 2), we mostly split the accessions alphabetically and added two Col-0 controls per imaging session. The rhizotrons of each imaging session were then swapped once between boxes to avoid rhizotron building effects. For the second imaging session (part 2), we made sure to include a Col-0 control to the bin that grew in the shelf outside of GLO-Bot (part 2, Bin D) to examine position effects. We now list the location of every accession in Figure 4—figure supplement 2. We added this information to the methods section on page 55.

We removed the comment about the block effects from the text since the final model did not take positional effects into account. It is true that there are more methods mixed into the results than usual. Given the “tools and resources” style of paper, we tried to include key pieces of methodology to help the reader understand the results.

8: So is it right that a subset of days are used? Why?

The six accessions demonstrate how GLO-bot can be used for continuous imaging of a small number of accessions with many replicates. With this in mind, the rhizotrons are imaged every day for a long period of time. Since the diversity panel had so many more accessions the plants were imaged for a subset of the days to account for possible hardware errors as they arose. We added this important information to the text on page 9.

8: PCA over time implies time dependent traits are just sized related.

We agree with this and added in “and plant size”.

9: ‘fitted values’ could be clarified.

We agree with the reviewer and have clarified the language regarding the fitted values. The description of fitted value calculation on page 9 has been revised to explain how the values were calculated and the manuscript has been reviewed to use consistent terminology. Figures have also been updated to indicate whether fitted values or raw values were used.

10: ‘published bioclimatic variables’ could include for clarity ‘…from the origin sites of the respective accessions.’

We appreciate this suggestion and have added the phrase in for clarity.

11: ‘progress in machine-learning’ – I remember I think one of the original rationales for this system was to image on both sides for a more complete root system? Was that in the original paper? Is that done here? With regards to bioluminescence – is it still clear that this method is advantageous to RGB imaging or other forms of imaging? For example, the Smith et al. 2020 paper shows many examples of identifying roots from complex backgrounds. The RootPainter software has been used for a rhizobox study:https://www.biorxiv.org/content/10.1101/2021.08.24.457510v1

The reviewer is correct that imaging from both sides of the rhizotron is an important aspect of the GLO-Roots/GLO-bot methodology. This imaging itself in the GLO-Bot system is performed in the same manner as the original paper, so imaging from both sides does indeed happen here and continues to be advantageous for extensively capturing the entire root system. Given the size of Arabidopsis roots, the bioluminescence continues to be advantageous. While the field of machine-learning for identifying roots is rapidly progressing, advances seem to be primarily focused on plant systems with much thicker roots. To our knowledge, RootPainter is still not sufficient for observing Arabidopsis roots in RGB images. We have added the comparison to RGB imaging to the discussion on page 13.

Can the possible advantages be discussed, especially if there are specific examples from images that show color imaging alone is not sufficient?

In the original GLO-Roots paper (Rellan-Alvarez et al., 2015), we scanned the rhizotrons in order to measure the moisture level of the soil. Since Arabidopsis has very thin roots, RGB imaging will not reveal the whole root system (see for example Ogura et al., 2019, https://www.cell.com/cell/pdf/S0092-8674(19)30684-1.pdf). Furthermore, manual tracing is required since the contrast between the roots and the soil is not strong enough. This approach would not allow the throughput and time-lapse aspect that we were aiming for. The advantage of luminescence is that it makes fine roots appear thicker, that it shines through thin layers of soil and that it improves the contrast between root and soil signal. We used Arabidopsis because of its big collection of natural variation, genetic resources, plant size and generation time in order to possibly identify genes that are involved in root system architecture growth. We aimed at visualizing development from young root systems to mature root systems to observe if root traits like average angle change over time. For species with bigger roots, however, RootPainter would be a very interesting pipeline to use and would save the step of plant transformation, reduce imaging time and image processing. We added RGB imaging as a possibility for bigger root systems to the discussion (page 13).

11: sample sizes could be sample numbers I think.

We agree with the reviewer and changed the text accordingly.

Figure 1 – scale needed in all panels.

We have added scale bars to all relevant parts of this figure.

48: Not clear why Materials and methods are in supplemental?

We thank the reviewer for this comment. Since this manuscript is not layouted yet, we decided that the figures and supplemental figures and tables should follow the main text. In the layouted version, however, the figures, the figure supplements and supplementary files will be included in the text so that the Materials and methods would follow the discussion. We agree that this is important since the methods section is essential, especially in an article that describes new methods.

51: "Accessions were always planted in the same locations relative to each other so the population was treated identically in each replicate." Strictly speaking – this is a bad choice, somehow you decided to intentionally NOT randomize which goes against statistical training and a strict reviewer may say it compromises the experimental design where all the measurements could be artifacts of position. Is there a good reason to not randomize? It seems odd in what otherwise seems like such careful design.

We thank reviewer #3 for this very valid criticism. Before we started imaging the GLO-Roots diversity panel, we carefully discussed possible strategies. We discussed if all 6 replicates should be imaged within one experiment (imaging run) or if all 93 accessions should be included. Furthermore, we discussed if we should randomize the positions of rhizotrons or if we should keep the same order. We decided for the latter to be able to examine positional effects. Since we included 4 Col-0 controls distributed among the 8 rhizotron boxes, we believed that we could examine positional effects more efficiently. Rhizotron positions for the diversity panel are mapped out in the new Figure 4—figure supplement 2.

51: 'four raw images' – root and shoot from each side is four? Maybe make clear.

Within the imager, there are two cameras on top of one another, taking an image of the top part of the rhizotron and one of the bottom part of the rhizotron. After those images were taken, the stage turns the rhizotron and the cameras take another top and bottom image, yielding 4 images per rhizotron. This was clarified in the text on page 55.

51: what is the 'correct blank images' – is this process entirely automated? Sounds like a user walks the processing through several fairly automated steps but not quite automated from image to data it seems.

This assumption is correct. One blank image was taken for each replicate. The blank image for the respective replicate is manually selected and then the automated script runs and subtracts the inherent background noise. This was clarified in the text on page 55.